# A stem cell population at the anorectal junction maintains homeostasis and participates in tissue regeneration

Louciné Mitoyan[1], Véronique Chevrier[1], Hector Hernandez-Vargas [2,3], Alexane Ollivier[1], Zeinab Homayed[4], Julie Pannequin[4], Flora Poizat[5], Cécile De Biasi-Cador[5], Emmanuelle Charafe-Jauffret [1,5], Christophe Ginestier [1] & Géraldine Guasch [1✉]

At numerous locations of the body, transition zones are localized at the crossroad between two types of epithelium and are frequently associated with neoplasia involving both type of tissues. These transition zones contain cells expressing markers of adult stem cells that can be the target of early transformation. The mere fact that transition zone cells can merge different architecture with separate functions implies for a unique plasticity that these cells must display in steady state. However, their roles during tissue regeneration in normal and injured state remain unknown. Here, by using in vivo lineage tracing, single-cell transcriptomics, computational modeling and a three-dimensional organoid culture system of transition zone cells, we identify a population of Krt17+ basal cells with multipotent properties at the squamo-columnar anorectal junction that maintain a squamous epithelium during normal homeostasis and can participate in the repair of a glandular epithelium following tissue injury.

---

[1] Aix-Marseille University, CNRS, INSERM, Institut Paoli-Calmettes, CRCM, Epithelial Stem Cells and Cancer Team, Marseille, France. [2] Department of Immunity, Virus and Inflammation, Cancer Research Center of Lyon (CRCL), Inserm U 1052, CNRS UMR 5286, Université de Lyon, Centre Léon Bérard, Lyon Cedex 08, France. [3] Department of Translational Research and Innovation, Centre Léon Bérard, Lyon Cedex 08, France. [4] CNRS, UMR5203, Inserm U661, Institut de Génomique Fonctionnelle, Montpellier, France. [5] Department of Biopathology, Institut Paoli-Calmettes, Marseille, France. ✉email: geraldine.guasch-grangeon@inserm.fr

Several of our organs have a region called transition zone (TZ) that connects two distinct epithelia such as stratified and glandular epithelia[1]. These junction areas can be found throughout the body in the eye, esophagus, stomach, ovary, cervix, and anus and are a site of associated pathology in human and mouse including cancers with poor prognosis[2–8] and pre-lesional conditions such as ulcer, Barrett's esophagus[9], and developmental disorders such as cloaca malformation[10]. These TZ are composed of cells expressing markers of adult stem cells[11–13] that can be the target of early transformation[14–17]. These junctions between our organs allow us to have a continuous lining of tissues, such as in the digestive tract. When affected, the anorectal region is usually associated with high invalidity of patients including fecal incontinence. It is thus essential to understand how important these TZ are to preserve organ homeostasis. In this work, by studying the properties of the anorectal TZ at the single cell level and by combining technologies such as organoid culture, gene sequencing and the use of mouse models, we have identified a population of Keratin 17 positive (Krt17+) basal cells with multipotent properties that maintain a squamous epithelium during normal homeostasis, and can participate to the repair of a wounded glandular epithelium.

## Results

**Multilineage potential of Krt17+ TZ cells during homeostasis.** In any species, the abrupt transition from one epithelia to another can be readily detected histologically and molecularly as each epithelium expresses characteristic proteins, such as Krt14/5 for the anal stratified epithelium and Krt8 for the glandular rectal epithelium[11] (Supplementary Fig. 1a and Fig. 1a). The hyperproliferative-associated-Krt17[18] marks specifically TZ cells[1,2] (Fig. 1a) with a low expression level in anal canal cells proximal to the TZ (Supplementary Fig. 1b). Krt17 expression correlates with the active renewal occurring in the anorectal TZ (Supplementary Fig. 1c). The mere fact that TZ cells can merge these different epithelia with separate functions implies for a plasticity, that these cells must display in steady state. However, it is unknown if TZ cells have any role in renewing of the epithelium during normal homeostasis and after an injured condition.

Inducible genetic fate mapping method allows the identification of stem cells in many epithelial tissues[19]. To investigate the contribution of the Krt17+ basal TZ cells to the renewing of the stratified and/or glandular epithelium, we performed lineage tracing using $K17CreER^{T2};R26R^{GFP}$ bigenic mice (Fig. 1b) using one injection of tamoxifen to label permanently few cells at the TZ (Fig. 1c, d). No Krt17+GFP+ cells were detected in non-induced $K17CreER^{T2};R26R^{GFP}$ mice analyzed short and long chase ($n = 3$) (Fig. 1d and Supplementary Fig. 1d) confirming the tight regulation of $K17CreER$ activity in the basal TZ cells. Since lineage tracing is reporting transcriptional activity of the $Krt17$ gene, we report the location of $Krt17$ transcripts in the anorectal region via high quality in situ hybridization. We showed by RNAscope technology that $Krt17$ transcript was exclusively found in the anal TZ region, when compared to the anal canal and the rectum (Supplementary Fig. 2a, b and quantified in Supplementary Fig. 2c). Furthermore, tamoxifen injection itself does not change $Krt17$ transcript (Supplementary Fig. 2c), neither Krt17 endogenous protein expression level (Supplementary Fig. 3a) and does not induce proliferation of Krt17+ TZ cells (Supplementary Fig. 3b).

We found that lineage-marked GFP+ TZ cells are found in the stratified epithelium of the anal canal after short and long-term tracing and not in the Krt8+ glandular epithelium (at least three mice for each time point analyzed) (Fig. 1c–e). GFP positive cells are still found in mainly all stratified epithelium of the anal canal after 1-year lineage tracing (Fig. 1d) strongly indicating that all the Krt17+ TZ cells are long-term, self-renewing adult stem cells. Moreover, GFP+ lineage traced cells colocalized with differentiated epithelial cell type Krt10+ and Loricrin+ (Fig. 1f) showing the multilineage potential of the Krt17+ stem cells.

To analyze the hierarchical organization of the anorectal cells we have performed single-cell RNA sequencing (scRNA-seq) analysis[20,21] from purified FACS-sorted cells (Supplementary Fig. 4), and we obtained initially eight distinct clusters (Supplementary Fig. 5a). A Gene ontology analysis for molecular pathways illustrated that one cluster was specially enriched in signaling cascades involving lipid and steroid metabolism (Supplementary Fig. 5b). This cluster represents anal glands that are present below the anal TZ. These glands are positive for the Oil red O staining and we confirmed at the protein level that perilipin-2, highly expressed in this cluster by sc-RNA seq, marks indeed the anal glands (Supplementary Fig. 5c). Two other clusters identify fibroblasts (Supplementary Fig. 5d) and hair follicle (Supplementary Fig. 5e) with known genes[22], such as $Lhx2$ and $Lgr5$ expressed in the bulge of the hair follicle but not in the anal TZ (Supplementary Fig. 5f, g). These three clusters were further removed from our analysis to only focus on the epithelial anorectal cells (Fig. 2a). From this new analysis, four clusters were found. One cluster, enriched in $Epcam$, $Krt8$ and numerous mucins mRNA, identified the rectum cell population (Fig. 2a, b). Differentiated anal canal cells are found in the cluster enriched in $Krt10$ and $Gpc3$ genes (Fig. 2c). Anal canal cells, depending on their proximal or distal localization, express low level of mRNA $Krt17$ (Fig. 2d). Notably, the anal TZ population can be separated into two $Krt17+$ clusters representing the basal and suprabasal cell layer (Fig. 2d, e). Glandular and stratified basal cells are marked by the epithelial polarized marker α6-integrin (Fig. 2e, f). Cells localized at the basal layer of the TZ show heterogeneity and can express differential level of Krt6 expression depending on their location. TZ cells closer to the anal canal express Krt6 only in the suprabasal layer (Fig. 2fi), whereas TZ cells closer to the rectum express Krt6 in the basal and suprabasal layers (Fig. 2fii). At this location, closer to the rectum, basal TZ cells also express the stem cell marker CD34[11] (Fig. 2g). The adherens junction protein nectin-4[23] marks the suprabasal TZ cells (Fig. 2e, f). The scRNA-seq data also shows that the glycoprotein Gpc3 seems to be exclusively expressed in the anal canal population compared to the TZ (Fig. 2c, h). Overall, the scRNA-seq data revealed heterogeneity within the anal TZ population (Fig. 2i), and suggest that cells located at different positions in the basal layer have distinct molecular signatures.

**Krt17+ TZ cells self-renew and differentiate in 3D organoid.** To test the long-term proliferation potential and the self-renewal ability of the anal Krt17+ TZ cells compared to the anal epithelium, we have used three-dimensional organoid culture system. We have used microdissection technique, mouse genetic and flow cell sorting using the $K17CreER^{T2};R26R^{GFP}$ mouse model induced with tamoxifen for 2 days to label specifically the TZ with GFP in combination with the cell surface marker Epcam expressed in a majority of epithelial cells (Fig. 3a and Supplementary Fig. 4). As a control we verified that tracing was initiated only in K17+ TZ cells; In vitro tamoxifen assay, where culture media was supplemented with tamoxifen for 1 week, confirmed that GFP was only induced in TZ organoids and not in anal canal organoids or with oil control supplementation (Supplementary Fig. 6a). Each sorted cell population, stratified (GFP-Epcam$^{Low}$), TZ (GFP+Epcam$^{Low}$), and glandular (GFP-Epcam$^{High}$) can grow as organoid culture and resemble their tissue of origin by histology (Fig. 3b), where stratified squamous epithelium in the TZ

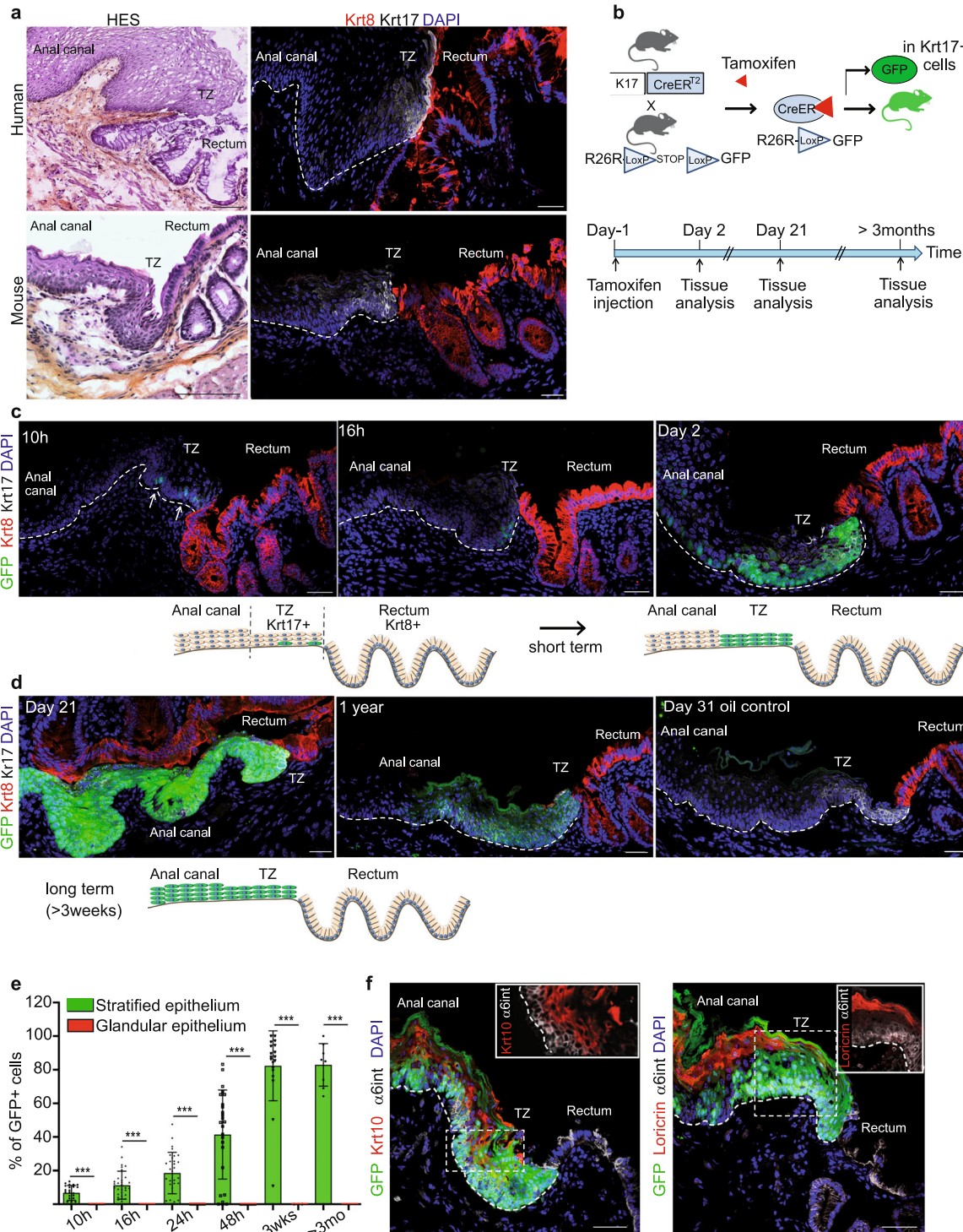

**Fig. 1 A population of basal Keratin 17+ cells at the anorectal junction contributes to the anal epithelium during normal homeostasis. a** Human and mouse anorectal TZ are histologically (HES) and molecularly similar. A population of keratin 17 (Krt17) expressing cells (white) shown by immunofluorescence is present at the anorectal TZ (*n* = 55 independent experiments from 58 mice and *n* = 3 independent experiments from two independent human samples). The rectum expresses keratin 8 (Krt8) (red). Scale bars for HES are 100 and 50 μm for immunofluorescence panels. **b** Schematic of the mouse model used and time point analyzed after one injection of 2 mg of tamoxifen to only label few cells at the TZ. **c, d** Unipotency of Krt17+ TZ cells (green) during short and long-term lineage tracing. Representative images of *n* = 3 biologically independent samples at each time point are shown. Dashed lines delineate the epithelium from the stroma. Scale bars are 50 μm. **e** Quantification of the % of GFP cells in stratified and glandular tissue (*n* = 3 mice per time point) (Source data are provided as a source data file). Two-tailed paired *t*-test; error bars, mean ± SD ***p < 0.0001. **f** Krt17+ TZ cells give rise to differentiated suprabasal layer cells of the anal canal (4 weeks post-tamoxifen injection) positive for Krt10 (red) and to cornified layer cells positive for Loricrin (red). α6 integrin (in white) marks the basal layer. Scale bars are 50 μm; Insets for Krt10 and Loricrin are zoom in 1.5-fold and 1.1-fold, respectively.

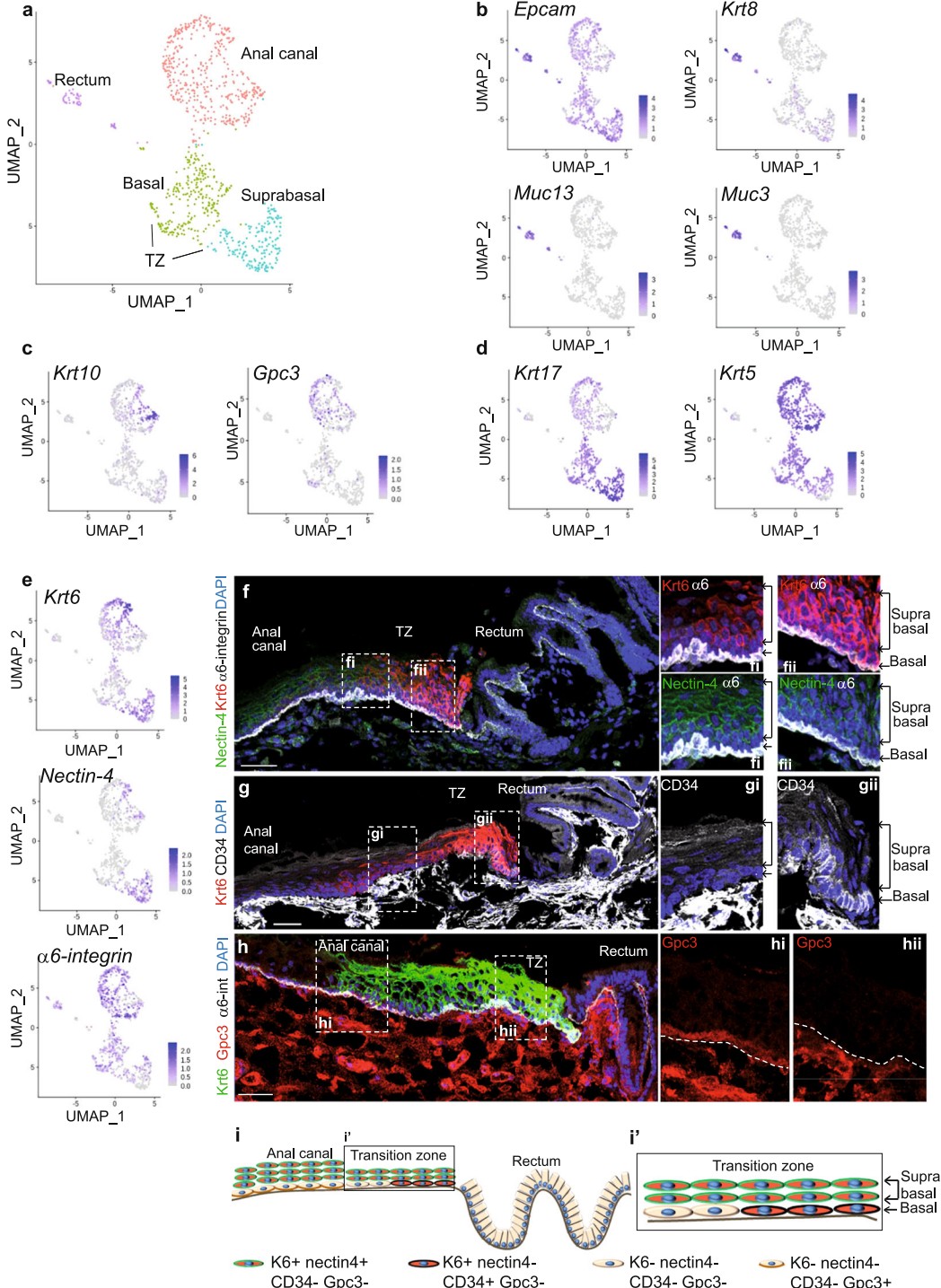

**Fig. 2 Single cell analysis reveals hierarchical organization of the anal TZ cells. a** Main cell populations visualized on Uniform Manifold Approximation and Projection (UMAP) dimensional reduction from scRNA sequencing of anorectal epithelial cells. Nine hundred and thirty-nine cells are visualized and colored by clustering. **b–d** Expression of key genes expressed in rectum, differentiated anal cells and TZ populations respectively. Darker color in the UMAP plot indicates higher expression level of the selected gene. **e** Expression of *Krt6* and *nectin-4* genes defines the suprabasal TZ population. **f** Immunofluorescence with Krt6 (red), nectin-4 (green) antibodies confirms that TZ population can be separated into basal (expressing the polarized α6-integrin in white) and suprabasal clusters ($n = 3$ independent experiments from three mice). Scale bar is 50 μm. **fi, fii** are zoom 2,5-fold into the TZ closed to the anal canal and the rectum respectively. **g** Immunofluorescence with Krt6 (red), and CD34 (white) shows that TZ basal cells closed to the rectum express the stem cell marker CD34 ($n = 3$ independent experiments from three mice). Scale bar is 50 μm. Boxed areas **gi** and **gii** are shown at higher magnification (2.5-fold). **h** Immunofluorescence with Gpc3 (red), Krt6 (green) and α6-integrin (white) shows the specific expression of Gpc3 in basal cells of the anal canal ($n = 3$ independent experiments from three mice). Scale bar is 50 μm. Boxed areas **hi** and **hii** are shown at higher magnification (2.5-fold). **i** Representation of the TZ heterogeneity depending on cell positioning. **i'** Zoom into the TZ.

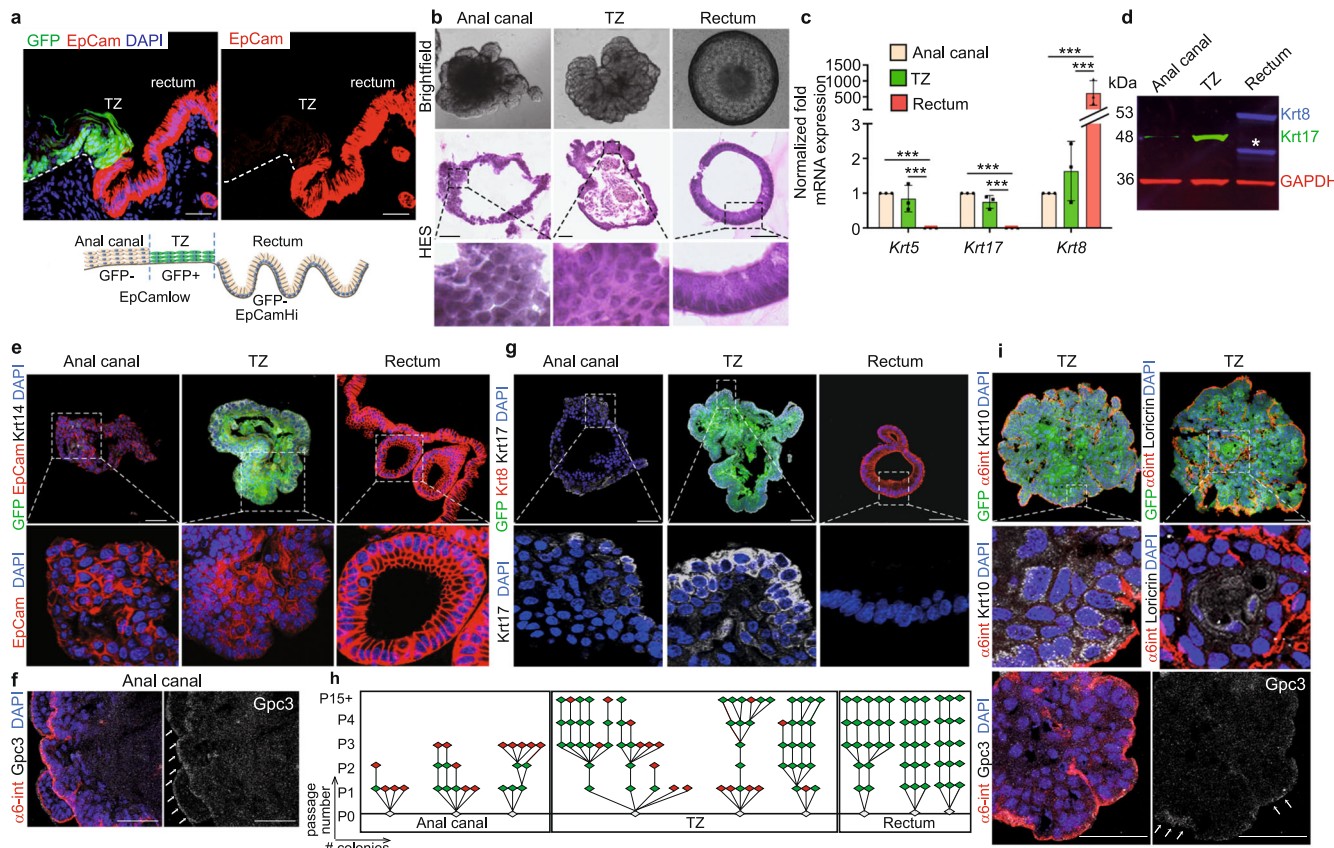

**Fig. 3 Keratin 17+ basal TZ cells are clonogenic in 3D organoid culture and have long-term self-renewal and differentiation potential. a** TZ cells are separated based on GFP and Epcam expression (red) ($n = 3$ independent experiments from five mice). See complete FACS strategy in Supplementary Fig. 4. **b** Brightfield and HES images showed the unique characteristics of the anal canal (squamous), anal TZ (squamous) and rectum (columnar) organoids. Representative images of $n = 21$ independent organoid culture experiments derived from $n = 21$ independent FACS sorting isolation after 14, 12, and 11 days of culture without passage for the anal canal, TZ and rectum respectively. **c** Quantitative qPCR analysis of *Krt5, Krt17, and Krt8* genes on growing organoids at passage 1 show the glandular nature of the rectum organoid expressing higher *Krt8* mRNA expression level than the anal canal and TZ ($n = 3$) (Source data are provided as a source data file). ***$p < 0.001$ calculated using two-way Anova and Bonferroni test; error bars, mean ± SEM. **d** Western Blot analysis of Krt17 (green) and Krt8 (red) showing higher expression of Krt17 in TZ organoids and Krt8 expression only in rectum organoids ($n = 3$ independent experiments from three independent FACS sorting) (Source data are provided as a source data file). Asterisk (*) denotes degradation of Krt8. GAPDH was used as a loading control. **e–g** immunofluorescences with various antibodies on organoids at passage 1 derived from FACS-sorted anal canal, TZ, and rectum cells showing that the molecular characteristics of the tissue of origin are maintained in culture organoids derived from the three cell populations. Anal canal and TZ organoids express the stratified-specific marker Krt14 (white), lower level of Epcam (red) compared to the rectum ($n = 4$ independent experiments from four biologically independent samples) (**e**), Anal canal organoids express Gpc3 ($n = 3$ independent experiments from three independent organoids) (**f**), whereas TZ organoids express high level of Krt17 expression (white) but no Krt8 (red) ($n = 3$ independent experiments from three independent organoids) (**g**). Images have been taken at the same exposure time and with the same laser intensity for each antibody for comparison. **h** Clonogenicity assay shows that anal canal-derived organoids have limited clonogenic property, whereas TZ and rectum-derived organoids survived 15 or plus passages ($n = 3$). Each triangle represents a single organoid. Red and green triangles respectively mean no growth and growth. P0 = No passage, P1 = passage 1. P2 = passage 2 etc.... **i** Immunostainings of differentiation markers Krt10 and loricrin (white) (respectively $n = 4$ and $n = 3$ independent experiments from three independent organoids) in TZ-derived organoid. α6-integrin (red) denotes the basal layer. TZ organoids can give rise to anal canal epithelial cells that express specifically Gpc3 (white) ($n = 3$ independent experiments from three independent organoids). Arrows show positive staining for Gpc3. Scale bars are 50 μm in **a**, **b**, **e**, **f**, **h**, and **i**. Insets for anal canal and rectum organoids are zoom in 3-fold and TZ organoid is zoom in 2-fold in **b**, 3-fold in **e**, 4-fold in **g**, 6-fold in **i**.

and anal canal organoids can be seen in contrast to a single layer of columnar cells in the glandular rectal organoid, molecularly by qPCR (Fig. 3c), Western blot (Fig. 3d), and immunofluorescence (Fig. 3e–g). The level of protein expression seen in the tissue is maintained in each organoid population such as Epcam^low^/Krt14+ in anal canal and TZ (Fig. 3e), Gpc3 in anal canal (Fig. 3f), Epcam^High^/Krt8+ in rectum (Fig. 3e–g) and Krt17^high^ in TZ organoid only (Fig. 3d, g). However, after subcloning and serial passages (Fig. 3h and Supplementary Fig. 6b), only the TZ organoids and not the anal canal can be maintained unlimited

time in culture (15 passages and plus which correspond to 131 days) demonstrating their ability to self-renew and validate their stem cell potential (Fig. 3h). As expected, rectum organoids can also be maintained unlimited time in culture (five passages and plus) as stem cells may reside in crypt similar than in the colon[24]. Because not all TZ organoid clones were successfully clonogenic, we tested if we could enrich for a basal TZ sub-population of cells by FACS-sorting cells based on Nectin4 expression (Supplementary Fig. 6c), as we previously showed their expression specifically in suprabasal layers of TZ (Fig. 2e, f).

Two populations were FACS-sorted; basal Nectin4− TZ cells and suprabasal Nectin4+ TZ cells (Supplementary Fig. 6c, d). Among these populations, only basal TZ population was able to be maintained for an unlimited time in culture compared to suprabasal TZ cells showing that only a small fraction of basal TZ displays stem cells properties (Supplementary Fig. 6d). The stem cell potential of the anal TZ is further confirmed by the expression of the differentiated markers Krt10 and loricrin expressed in the TZ-derived organoids (Fig. 3i), and the anal canal marker Gpc3 also found in some regions of the TZ organoid (Fig. 3i) showing the multilineage potential of the Krt17+ cells as previously demonstrated in vivo (Fig. 1f).

**In silico approach reveals anal TZ differentiation path**. Next, we have used computational models to predict the hypothetical trajectory or pseudotime[25] that could stand from our single cell mRNA sequencing datasets. We have examined the differentiation trajectory that could exist between the anal TZ clusters, the anal canal cluster and the rectum cluster. Differentiation trajectory reconstruction on the four remaining clusters using the slingshot R package[26,27] suggests three trajectories (Fig. 4a). However, only two trajectories suggest a differentiation trajectory. This is illustrated by pseudotime values, which contain information about ranking and distance of every cell within a given trajectory (Fig. 4b). As shown, Trajectory 1, which suggests a differentiation progression from the anal TZ basal cells to the anal canal (Fig. 4c), displays the longest distribution of pseudotime values, spanning more clusters and cells (Fig. 4b). Trajectory 2, which describes a hierarchical relationship between the anal TZ basal cells and the anal TZ suprabasal (Fig. 4c), shows also a wide range of pseudotime values, in contrast to trajectory 3 (rectum) in which all values are concentrated in a shorter range (Fig. 4b). This result indicates that trajectory 3 is essentially one cluster of relatively homogeneous cells, which doesn't suggest a differentiation trajectory. Altogether, the scRNA seq analysis supports our in vivo lineage tracing data and our in vitro organoid assay showing that, during normal homeostasis, anal TZ cells maintain the stratified squamous epithelium by giving rise to suprabasal TZ cells and anal canal population (Figs. 1 and 3h–i).

**Injured Krt17+ TZ cells display a phenotypic plasticity**. Another essential feature attributed to stem cells in vivo is their ability to repair injured tissue[28,29]. To test if the anorectal TZ cells were capable of epithelial reconstitution in vivo and investigate their potential plasticity, we lineage-traced TZ Krt17+ GFP+ cells after removal of adjacent rectal glandular epithelium (Fig. 5a, b). Four weeks is generally sufficient for the glandular epithelium to be repaired[30]. We therefore analysed the contribution of the GFP+-derived anal TZ cells to the repairing glandular epithelium at the beginning, and at the end of the healing phase (1 and 4 weeks of post-wound respectively). Interestingly we found that at 1 week of post-wound, few GFP+ derived anal TZ cells, continuous to the TZ, are found in a transition state by co-expressing the Krt17 and the Krt8 proteins (Fig. 5c, d) in three mice over five analysed. In contrast, at 4 weeks of post-wound we detected GFP+ rectal crypt derived from anal Krt17+ TZ cells in eight mice over 13 analysed (Fig. 5c, d). No GFP+ crypts were found in oil injected mice 4 weeks of post-EDTA wound. At least 100 slides with 9 μm-thick sections were screened for each mouse (n = 4) (Fig. 5c). These newly formed crypts expressed Krt8 but not Krt17 (Fig. 5c) and contain differentiated cell types, such as enteroendocrine cells (positive for chromogranin A) and alcian-blue positive-goblet cells specifically found in glandular epithelium demonstrating their ability to give rise to cells from multi-lineages (Fig. 5e). Anal Krt17+ TZ cells are therefore multipotent

in response to epithelial injury. The GFP+ rectal crypts derived from the Krt17+ TZ cells are not found continuous to the TZ such as seen, when colonic stem cells are lineage traced after injury[31]. We postulate that the wound activates two stem cell compartments (the rectal stem cell and the anal TZ) and rectal cells are dominantly responsible for the regeneration. Moreover, staining with EdU at shorter time point (48 h) after the wound (Supplementary Fig. 3b) shows that rectal cells are highly proliferative compared to TZ cells. This suggests that residual rectal epithelial cells, closer to TZ, rapidly expand and could contribute to the distance between anal Krt17+ TZ cell tracing and TZ over time. To further confirm TZ cells multipotency, we also challenged TZ cells by performing a second type of injury. K17CreER[T2];R26R[GFP] mice were injected twice with tamoxifen to mark TZ cells and anal TZ was mechanically wounded the next day (Supplementary Fig. 7a, b). Similar to EDTA wound, at 1 week post-mechanical wound, few GFP+ anal TZ cells, continuous to the TZ, co-expressing Krt17 and Krt8 were found (Supplementary Fig. 7c). Furthermore, GFP+ cells were also co-localizing with alcian-blue positive goblet cells confirming their multipotent property (Supplementary Fig. 7c). Together, these findings establish that a minority of anal Krt17+ TZ cells, when challenged by removing its direct neighbors, can contribute to rectal repair by their ability to change their fate that allow them to give rise to a different epithelium in response to injury.

To investigate dynamics of Krt17+ cells in regeneration, we performed single cell analysis upon injury (Fig. 6a and Supplementary Fig. 8). K17CreER[T2];R26R[GFP] mice were injected twice with tamoxifen and subjected to EDTA wound the next day (Fig. 5a, b). At 1 week of post-EDTA wound, cells were sorted following the established FACS strategy based on Epcam expression, which was maintained (Supplementary Fig. 8a, b). After single-cell RNA sequencing, we obtained ten clusters (Supplementary Fig. 8c) including hair follicles, fibroblasts, and anal glands clusters, which were further removed as previously done (Supplementary Fig. 5) to focus on anorectal cells. After removal of these three clusters (Supplementary Fig. 8c, d), seven clusters were identified (Fig. 6a). Two clusters with stratified epithelial cells enriched with wound/regeneration genes, such as IL-33 were found (Supplementary Fig. 8e). One cluster enriched for differentiation markers such as Krt10 was identified as anal canal/TZ suprabasal cluster (Supplementary Fig. 8e). Two clusters enriched for rectum specific genes were found such as Muc13 and Krt8 (Fig. 6). Transition zone cells were identified in one cluster enriched in previously shown markers, such as Nectin 4 and Krt6a (Supplementary Fig. 8e and Fig. 2g). More interestingly, an additional cluster, absent in unwound analysis (Fig. 2), was identified as "TZ hybrid state" cells. This cluster is enriched for TZ marker Krt17 and glandular markers Muc13 and Krt8.

Then, pseudotime analysis was performed on the post-wound single cell RNAseq dataset (Fig. 6b). We found three major trajectories representing a hierarchical relationship between TZ cells to anal canal/suprabasal cells and wounded epithelial cells. The third trajectory stands for TZ cells transitioning to the "TZ hybrid state" to finally go towards the rectum which support our previous data, where TZ cells displayed multipotency when subjected to injury (Fig. 5). In contrast to the normal condition (Fig. 4), the rectal trajectory in the wounded tissue (pseudotime 3 Fig. 6b) displays a differentiation path. Although pseudotime values are somehow concentrated at the bottom, reflecting the homogeneous rectal cell cluster, their distribution is clearly more elongated than the rectal normal trajectory. This has been confirmed by performing an unsupervised pseudotime analysis on a subset of clusters including the TZ hybrid and rectal cells (Fig. 6c). One trajectory was found suggesting a hierarchical relationship between TZ Krt17+ cells and Krt8+/Muc13+ cells.

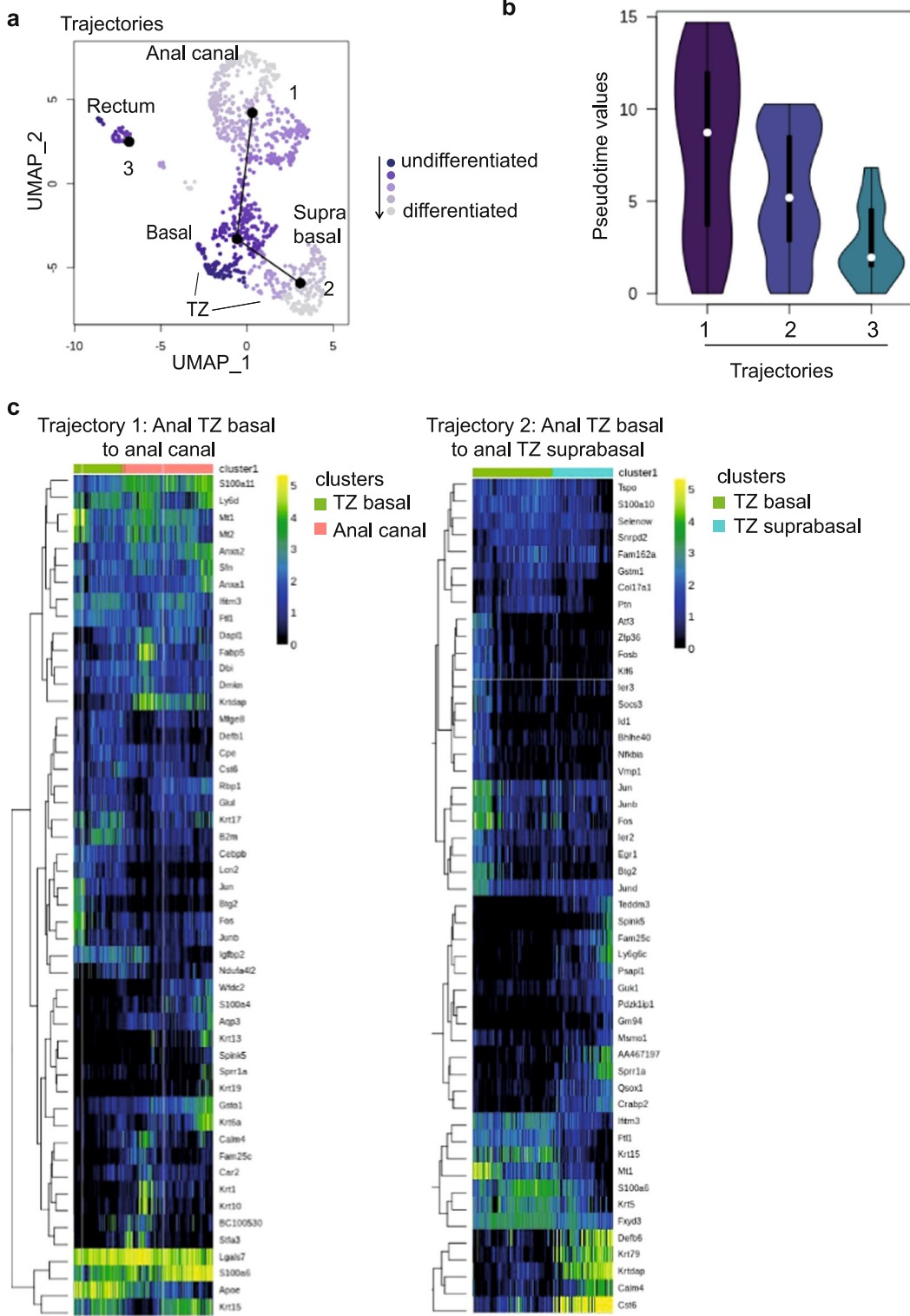

**Fig. 4 Pseudotime analysis reveals putative anal TZ differentiation trajectories. a** Structure of the differentiation trajectory reconstruction using the slingshot R package suggests three trajectories but only two trajectories reflect a differentiation path. Trajectory 1 represents a hierarchical relationship between anal TZ basal cells and anal canal cells and anal TZ basal cells to anal TZ suprabasal cells in trajectory 2. **b** The violin plot shows pseudotime values for all trajectories. The distribution of pseudotime values reflects the heterogeneity of the lineage. If there are more cells, there is a trend to have a longer plot. The round shape of the violin plot from trajectory 3 indicates that this is essentially one cluster and most cells are homogeneous, which do not suggest a differentiation trajectory in contrast to trajectory 1 and 2. **c** Clustering analysis showing the top genes significantly associated with the differentiation paths of trajectories 1 and 2. Heatmaps represent normalized gene expression in logarithmic scale.

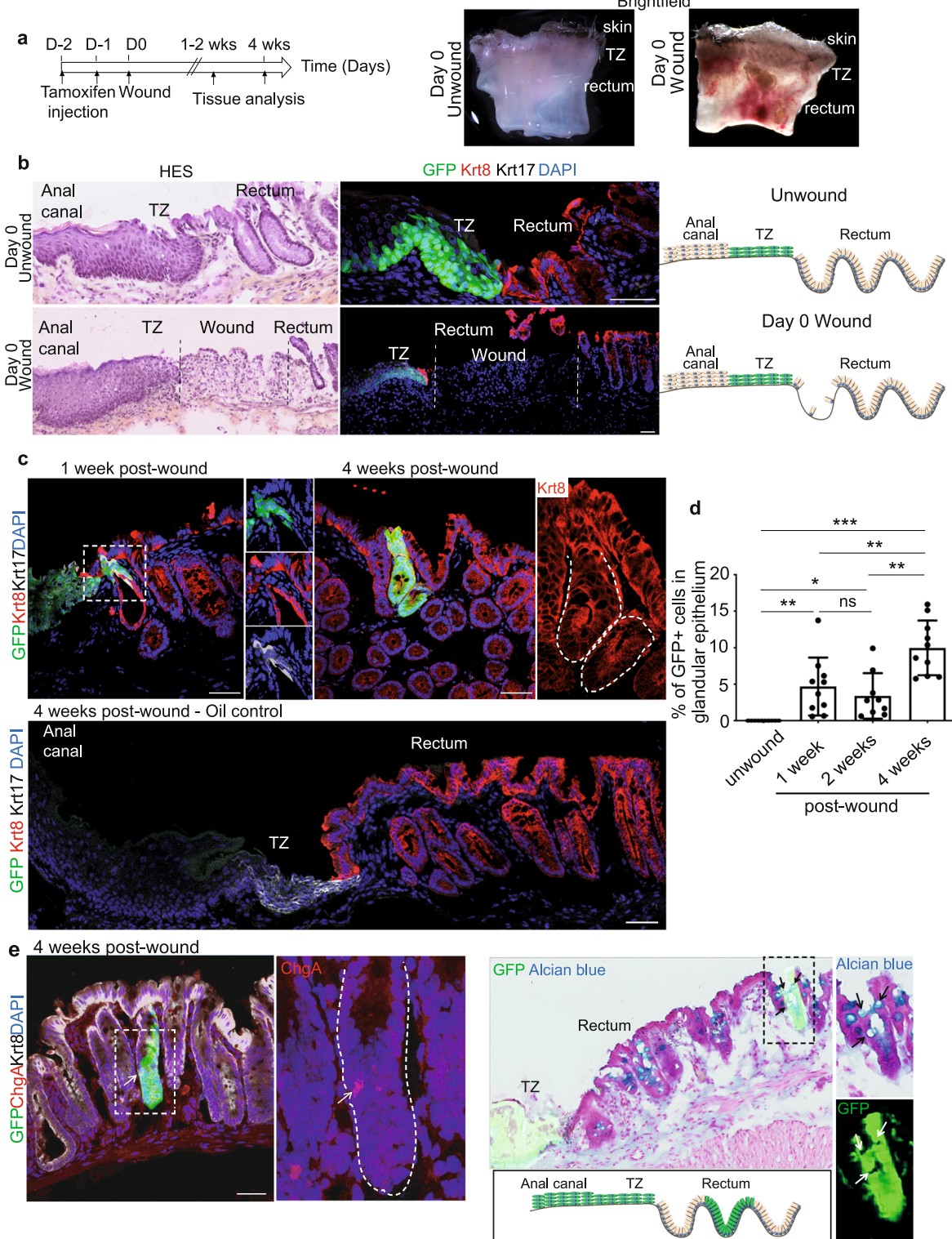

**JunB is involved in the proper differentiation of anal TZ**. We then analyzed pathways enriched in Krt17+ cell clusters and focused on the Jun family as interestingly this pathway has been involved in other epithelial stem cells, such as hair follicle bulge stem cells and is associated with wound response[32]. First, we have confirmed by immunostaining that the AP-1 transcription factor JunB was enriched at the anal TZ (Fig. 6d, e), but not the anal canal or the rectum confirming the scRNAseq analysis. Moreover,

in a wounded situation we showed that JunB is also expressed in cells participating in both chemical and mechanical wound repair (Fig. 6f and Ext. Data Fig. 7c). We further showed using shRNA in the organoid culture, that JunB is essential for TZ cells ability to differentiate properly. Two shRNA JunB (sh2 and sh3), which have the most efficient JunB downregulation, showed alteration in differentiation markers Krt10 and loricrin by immunofluorescence and western blot analysis on FACS sorted

**Fig. 5 Multipotency of anal TZ after a wound injury. a** Wound strategy in *K17CreER*$^{T2}$;*R26R*$^{GFP}$ bigenic mice induced with tamoxifen 2 days before injury. Brightfield images at day 0 shows the localization of the wound in the entire tissue. **b** Validation of the removal of the rectal crypt closed to the anal TZ by HES and immunostaining positive for Krt8 (red) (*n* = 3 independent experiments from three mice). **c** At 1 week of post EDTA wound, few TZ-derived GFP+ cells were found in 3/5 mice in the rectum area in a transition state co-expressing the epithelial glandular marker Krt8 (red) and the epithelial TZ marker Krt17 (white). Insets are zoom in 4-fold (left panel) and 2.5-fold (right panel). At 4 weeks post EDTA wound, GFP+ crypt derived from basal Krt17+ anal TZ cells were found in 8/13 mice showing their multipotency. Krt17+ anal TZ cells give rise to Krt8+ glandular cells (inset 2.5-fold zoom). Representative images of *n* = 4 biologically independent samples of oil control injected mice analyzed 4 weeks of post-EDTA wound and stained with Krt17 in white and Krt8 in red. At least 100 slides with 9 μm-thick sections were screened for each mouse and no GFP+ crypts were found. **d** Quantification of the percentage of GFP positive cells, representing the anal TZ Krt17+ derived cells, found in the glandular epithelium at 1week, 2 weeks and 4 weeks post-wound (*n* = 3 mice quantified for each time point). (Source data are provided as a source data file). Two-tailed paired *t*-test; error bars, mean ± SEM *$p$ = 0.0115, **$p$ ≤ 0.0100, ***$p$ < 0.0001. **e** Krt17+ anal TZ cells give rise to differentiated enteroendocrine cells (Chromogranine A+, inset 3-fold zoom) and goblet cells (alcian blue+, inset 2.5-fold zoom) (*n* = 3 independent experiments from three mice). Scale bars are 50 μm.

RFP+ organoids (Fig. 6i, j). These data suggest that JunB may participate into the wound repair by promoting differentiation of the anal TZ cells, which may undergo further plasticity program to give rise to a fully functional glandular cell type.

## Discussion

These results of in vivo lineage tracing, 3D organoid culture and in vivo regeneration assays using mouse TZ samples, identified a reservoir of multipotent stem cells for surrounding epithelia that participate to the maintenance of both tissue type. This study has global relevance to stem cell function in many other regenerative tissues. We showed that anorectal TZ cells, which are locally restricted between the stratified squamous epithelium of the anal canal and the simple glandular epithelium of the rectum, contributes to tissue renewal of the anal canal only upon normal homeostasis. Upon injury a minority of anal Krt17+ TZ cells can contribute to rectal repair and give rise to a fully differentiated rectal crypt. This cell fate change during their response to wounding is a common property of many epithelial stem cells in their tissue of origin[33]. In contrast to other stem cell niches, transition zones are naturally subjected to mechanical stress and microbiota exposition[1]. Thus wounding and healing mechanisms are often activated in the TZ. Here, anal TZ, which have initially squamous-like properties, are able to repair and become a fully differentiated glandular cell-type. This plasticity induced by injury of the surrounding epithelium, may also exist in the progeny of the anal TZ as reported in hair follicle bulge stem cells[34] and in the eye limbus TZ following ablation of the stem cell compartment[35]. One of the existing theories is that different epithelial tissues have their own unique tissue specific stem cells to maintain homeostasis of the tissue from which they are derived[36]. Instead, here we propose that TZ cells could represent a backup reservoir of stem cells for surrounding epithelium and their phenotypic plasticity seen during wound healing could represent an hallmark of tumor initiation[37]. In human and mouse, TZ are cancer-prone stem cell niches. Therefore, understanding how these TZ cells participate in driving cancer will be highly relevant for cancer prevention and regression.

## Methods

**Mice.** Mice are housed in a sterile barrier facility as previously described[38]. The housing conditions of all the mice followed strictly the ethical regulation. Mice are housed in individually ventilated cages (IVC, sealsafe plus TECHNIPLAST) according to SPF FELASA standards and food and water were given at libitum. For social enrichment, six mice per cages are generally housed with sterile nesting materials such as cotton or compressed wood chips for nidification. The room temperature ranged from 20 and 25 °C. The relative ambient humidity was 55% +/−15. Semi-natural light cycle of 12:12 was used. All experiments were approved by the European and national regulation (ethical committee C2EA14, protocols # 4572, #8287, and #2244) and carried out using standard procedures. Transgenic mouse lines have been previously described: *K17CreER*$^{T2}$[39], *Rosa26-lsl-ZsGreen1* (Jackson laboratories, stock no.007906). We have backcrossed our double

transgenic mouse lines into the C57/BL6J background for six generations. Females and males have been used in all experiments.

**Lineage tracing analysis.** *K17CreER*$^{T2}$ mice were crossed with *Rosa26-lsl-ZsGreen1* reporter mice, which express the Green Fluorescent protein following Cre excision of a *loxP*-flanked cassette, to generate *K17CreER*$^{T2}$;*R26R*$^{GFP}$ bigenic mice (Fig. 1b). To induce GFP expression, *K17CreER*$^{T2}$;*R26R*$^{GFP}$ mice received daily intraperitoneal injections of 2 mg of tamoxifen (Sigma) or sunflower oil alone (vehicle) for 2 days. Anorectal specimens (*n* = 3 mice per time point) were harvested at the following time after tamoxifen administration: 10 h, 48 h, 3 weeks, and 1 year (Fig. 1c, d). Percentage of GFP+ cells was quantified by analyzing at least ten different areas of anorectal TZ from 19 mice. Total number of cells were counted using DAPI staining in stratified and glandular epithelia. The number of GFP+ cells was counted using Zen software and photoshop CC2019 and the percentage of GFP cells was calculated according to total number of cells (reported as 100%).

**Immunostaining and histology.** The anorectal tissue was fixed in 4% paraformaldehyde overnight at 4 °C. The sample was then washed three times in PBS 1× and put in sucrose 30% overnight at 4 °C. Lastly, the tissue was put in two volumes of sucrose 30% and one volume of OCT (Tissue-Tek, Sakura, Torrance, CA) for few hours and embedded in OCT compound. For some antibodies, such as Krt6 and nectin-4, the anorectal tissue was fixed in methanol for 30 min at −20 °C instead of the paraformaldehyde and process in 30% sucrose, as previously described. Ten micrometer frozen sections were used to performed immunofluorescence. Briefly, sections were permeabilized in 0.1% Triton X-100 (Euromedex 2000-B) in PBS 1× for 10 min at room temperature and blocked in 2.5% normal goat serum, 2.5% normal donkey serum, 2% gelatin (Sigma G7765), 1% bovine serum albumin, 0.1% Triton X-100 in PBS 1× for at least 30 min at room temperature. Whenever mouse antibodies were required, the MOM kit (vector Lab) was used. Primary antibodies were incubated in the blocking solution overnight at 4 °C. After three washes for 10 min in 0.1% Triton X-100 PBS 1×, secondary antibodies diluted in the blocking solution containing DAPI (1 μg/ml Thermo-Scientific 62248) was added for 45 min at room temperature. After three washes for 10 min in 0.1% Triton X-100 PBS 1×, slides were mounted in prolong Gold antifade (Invitrogen, P36934). The list of primary and secondary antibodies is in the Supplementary Table 1. Fluorescent images were acquired using a Confocal microscope ZEISS LSM 880. All images were analyzed using ZEN software and Photoshop. Histologies were performed by the IPC/CRCM Experimental Pathology platform (ICEP), and analyzed by two independent pathologists at the Paoli Calmettes Institute.

*EdU staining.* Mice were injected intraperitoneally with 0.25 mg EdU (Abcam, ab146186) for 4 h and tissue was processed as described in the previous sections. The Click-iT EdU AlexaFluor 647 Imaging kit, (Invitrogen C10640) was used to detect proliferative cells according to the manufacturer instructions.

*Oil Red O staining.* Slides were dip in 60% isopropanol one time quickly then stained in working Oil Red O solution (30 ml of stock solution mixed with 20 ml of distilled water) for 15 min at RT. Slides were dip in 60% isopropanol one time quickly followed by one quick dip in distilled water. Slides were counterstained with EnVision™ Hematoxylin (SM806 Dako K8008) for 5 min and washed ten times in distilled water. Slides were mounted with Aqueous mounting gel (Aquatex Merck Milipore 1.08532.0050).

*Alcian blue staining.* OCT sections (9 μm) were stained with Alcian Blue (Biognost AB2-OT-100) according to the manufacturer's instructions and counterstained with nuclear fast red (Merck).

**RNAscope fluorescent assays.** To detect *Lgr5* and *Krt17* at the mRNA level, OCT sections (9 μm) were prepared on superfrost plus slides and RNAscope highly sensitive probes against *Lgr5* (Cat # 311021-C2; ACD Inc.), and *Krt17* (Cat #479911;

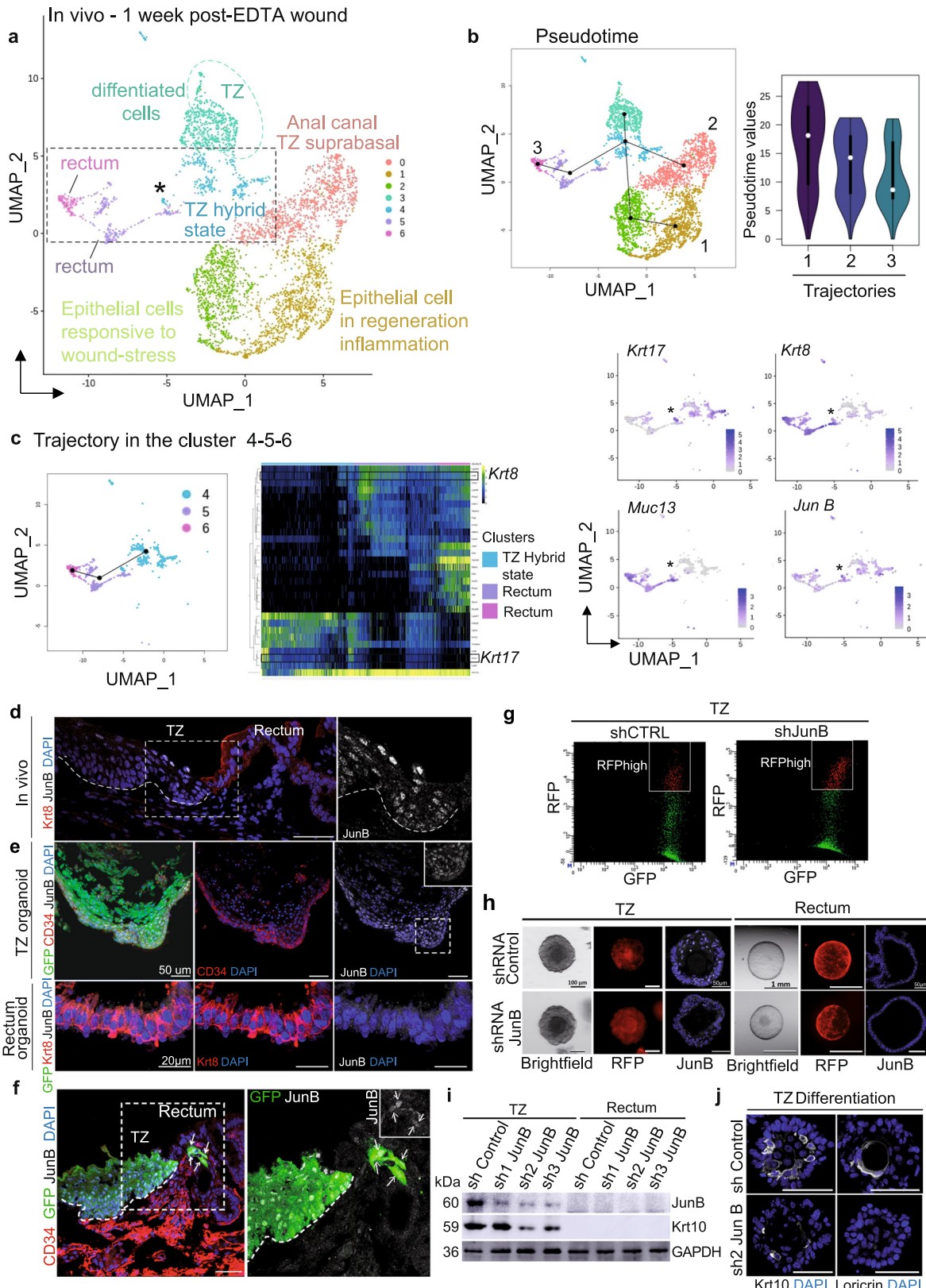

ACD Inc.) were used in the RNAscope fluorescent assay according to the manufacturer's protocol (Catalog Number 320293). Briefly, fixed frozen sections were baked at 60 °C and then post-fixed using 4% paraformaldehyde. Gradient dehydration using 50%, 70 and 100% ethanol was performed and slides were then processed for antigen retrieval followed by pretreatment with protease III, and then incubated with targeting probes for 2 h at 40 °C in a humidity-controlled chamber. Finally, AMP signal amplification followed by RNAscope Amp4-AltB(550) was carried out for fluorescent visualization. A housekeeping gene *POLR2A* was used as a positive control to ensure the RNA quality of the tissue, in addition, a negative control probe targeting bacterial *DapB* was used to assess nonspecific background labeling.

**Human samples**. Anorectal tissue samples from adult patients were approved by the Paoli-Calmettes Institute Strategic Orientation Committee (authorization

**Fig. 6 Post-wound single cell RNA sequencing reveals a TZ population enriched in the transcription factor JunB. a** Main cell populations visualized on UMAP from scRNA-seq of anal epithelial cells 1 week of post-EDTA wound **b** Structure of the differentiation trajectory reconstruction using the slingshot R package suggests three trajectories. Pseudotime values show an elongated distribution on each trajectory suggesting that there are more intermediate phenotypes and therefore a differentiation process. **c** Hierarchical relationship between clusters 4, 5, and 6 and expression of key genes expressed in TZ hybrid state cells. **d** In tissue, JunB (white) is specifically expressed by transition zone cells and is absent in Krt8 (red) glandular epithelium ($n = 3$ independent experiments from three mice). **e** Similar to in vivo, CD34 (red) and JunB (white) expressions are specific to TZ-derived organoids but not organoids from the rectum ($n = 3$ independent experiments from three TZ and rectum organoids). **f** One week of post-EDTA wound, few GFP+ (green) cells participating in the regenerative crypt are JunB+ (white) ($n = 3$ independent experiments from three mice). Insets are zoom in 1.7-fold. **g** TZ organoids infected with control shRNA-RFP and shJunB-RFP were FACS-sorted based on double expression of GFP and RFP ($n = 3$ independent FACS from three independent samples). **h** Brightfield and RFP images of TZ and rectum organoids infected with control shRNA or shJunB (red). JunB staining (white) revealed downregulation in TZ organoids infected with shJunB compared to control shRNA. Rectum organoids were used as negative control ($n = 4$ independent experiments from four independent samples). **i, j** *JunB* knockdown impairs TZ cells differentiation properties. **i** Western Blot analysis showing JunB downregulation in TZ organoids infected with sh2 and sh3 JunB compared to control shRNA and sh1JunB infections ($n = 6$ independent experiments from six biologically independent samples). Differentiation marker Krt10 expression is decreased following JunB downregulation in organoids infected with sh2 and sh3 JunB ($n = 2$ independent experiments from two biologically independent samples). GAPDH was used as a loading control. (Source data are provided as a source data file). **j** Immunofluorescence confirms the downregulation of Krt10 in sh2 and sh3 JunB organoids ($n = 5$ independent experiments from 5 independent samples). The defect in differentiation is also detected using antibody against Loricrin ($n = 5$ independent experiments from five independent samples). Scale bars are 50 μm (**d**, **e**) (TZ organoid) (**f**, **j**), and 20 μm (**e**) (TZ rectum).

---

TZ-cancers-IPC 2015-021). Informed written consent was given by the patients. Samples came from discarded tissues after surgery.

**Isolation and purification of anorectal epithelial cells by flow cytometry**. The anorectal regions of at least five $K17CreER^{T2};R26R^{GFP}$ mice induced with tamoxifen for 2 days were microdissected under a dissecting scope (Leica MZ6), and pooled in epithelial cell culture medium[40] containing 10% fetal bovine serum (FBS) for further dissociation. Tissues were cut in small pieces using scissor and scalpel in 4 ml HBSS 1× + 50 μl collagenase 20% from *Clostridium histolyticum* (Sigma C2674) and were incubated with high agitation 45 min at 37 °C. Then, 4.2 μl of DNAse I (10 mg/ml Sigma DN25) were added followed by 10 min incubation at 37 °C under agitation. Samples were then resuspended with 12 ml cold PBS 1× and placed in a 50 ml conical tube filled until 50 ml with PBS 1× and centrifuged for 10 min at 4 °C at 200×g. Pellets were resuspended with 20 ml cold PBS 1× and filtered through 70 μm filter. To avoid cell death, 200 μl of FBS was added. To dissociate left over tissues on the filter, 1 ml of TrypLE express enzyme (Gibco™, 12605010) was added followed by 5 min incubation at 37 °C. Trypsin activity was then stopped by adding 3 ml of epithelial cell culture medium +10% FBS and the mix was filtered through 40 μm filter and centrifuged 10 min at 4 °C at 300×g. Pellets were resuspended in PBS 1 × 2%FBS. Primary antibody was added to samples 30–45 min on ice and covered with foil. Samples were then washed with PBS 1× 2% FBS and centrifuged 5 min at 4 °C at 300×g. After discarding the supernatant, secondary antibody was added for 20 min at 4 °C and then washed again with PBS 2% FBS. Finally, FVD780 (1/1000, Thermofisher) was added to stain dying cells. Cell sorting was performed using a FACS Aria II equipment (BD Bioscience) and FACS diva software.

**Organoid culture**. After FACS isolation, 4000–10,000 cells were cultured in 25 μl Matrigel (Corning, 356231) on 48-well plate. Three hundred microliter of medium (Supplemental Table 2) supplemented with 10 μM Y-27632 (only first 5 days to avoid anoïkis) were added to wells and the plate was incubated at 37 °C, 5% CO₂. The medium was changed every 3–4 days. Upon outgrowth, organoids were passaged every 10–15 days either by mechanical dissociation by gentle pipetting or by enzymatic dissociation with TrypLE express enzyme (Gibco™, 12605010). Medium was supplemented with 10 μM Y-27632 the first 3 days.

*Imaging organoids*. Brightfield images, GFP and RFP expressions were monitored using the Leica M205 FA stereomicroscope.

*Embedding for histology*. Organoids were washed with PBS and incubated for 30 min at 37 °C with 300 μl of low melting agarose 2% (Promega, V2111). Polymerized agarose was transferred to a 1.5 ml Eppendorf and fixed in 4% paraformaldehyde overnight at 4 °C under agitation. The sample was then washed three times in PBS 1× and put in PBS sucrose 30% overnight at 4 °C. Finally, organoids were incubated with gelatin 7.5%/sucrose 15% in PBS at 37 °C for at least 30 min and put in mold to freeze at −80 °C.

**Clonogenicity assay**. Organoids were washed with PBS and incubated for 5 min at 37 °C with 200 μl of medium (Supplemental Table 2) + 1 μl collagenase 20% from Clostridium histolyticum (Sigma C2674) to dissociate matrigel. Under the stereomicroscope (Leica M205 FA), for each passage, at least three organoids of around 150 μm were isolated for anal canal and TZ and 540 μm for rectum. The size difference corresponds to the fact that stratified organoids (anal canal and TZ)

contain more cells overall than simple glandular epithelium organoids (rectum). Each organoid was dissociated for 5 min at 37 °C with 20 μl of TrypLE express enzyme (Gibco™, 12605010) and after washing, replated in 25 μl of matrigel. Three hundred microliter of medium (Supplemental Table 2) supplemented with 10 μM Y-27632 (only the first 5 days to avoid anoïkis) were added to wells and the plate was incubated at 37 °C, 5% CO₂.

*In vitro tamoxifen assay*. Anal canal cells were sorted by previously established FACS strategy, cultured in 25 μl Matrigel (Corning, 356231) on 48-well plate and incubated at 37 °C, 5% CO₂. At passage 1, culture media was supplemented with oil control or tamoxifen (0.1 μM) in culture media during 7 days and the medium was changed three times every 2 days. As a positive control, we sorted TZ cells based on Epcam expression from $K17CreER^{T2};R26R^{GFP}$ bigenic mice non-induced with tamoxifen, and we added the tamoxifen in culture media similarly to the anal canal.

*Lentiviral infection of organoids with ShRNA JunB*. Vector control (SMARTvector Non-Targeting Mcmv-TurboRFP control particles #S08-005000-01) and sh JunB 1,2,3 vectors (sh1 JunB #V3SM7592-234927818, Sh2 JunB #V3SM7592-236055786 and sh3 JunB #V3SM7592-237743179) were produced by Horizon. $2.5 × 10^6$ viral particles were combined with 2 μg of polybrene (Sigma TR1003-G) and incubated 10 min at room temperature in organoid culture medium. $2 × 10^4$ dissociated organoids cells were then added to virus and polybrene solution and incubated 10 min at room temperature. Twenty-five microliter of Matrigel (Corning ref 356231) were added to the mix and 25 μl of the suspension were dispensed into the center of each well of a 48-well plate. After 10 min, 300 μl of medium containing Y-27632 was added. Twenty-four hours after infection organoids were washed by adding 10 ml of basal medium and centrifuged as previously described. The pellet was suspended with Matrigel and suspension was dispended in wells as previously described. Y-27632 was added to the medium the first 3 days.

**Western blot**. Organoids were lysed with lysis buffer (50 mM Tris, pH7.2; 350 mM NaCl; 1% Triton X-100; 0.5% Na deoxycholate; 0.1% SDS; 20 mM MgCl₂ and Inhibitor Cocktail) and proteins were separated by NuPAGE 4–12% Bis–Tris 1.0 mm mini protein gel, transferred to nitrocellulose membranes and blocked for 1 h with 5% non-fat milk or 5% BSA in TBS 1× (Tris Buffer Saline: 1 M Tris pH 7.4–7.5; 5 M NaCl) depending of the primary antibody. Membranes were subjected to immunoblotting using antibodies against the following proteins at the indicated dilutions: GAPDH (1/5000), Krt17 (1/20000), Krt8 (1/4000), Krt10 (1/500) in 5% non-fat milk and Jun B (1/2000) in 5% BSA. See Supplemental Table 1 for complete reference list of the antibodies used. Full scan blots are provided in the Supplementary information. Membranes were then washed three times with TBS 0.1% Tween for 10 min. Fluorescent secondary antibodies or HRP-coupled secondary antibodies listed in the supplementary Table 1 were used at 1/1000 in 5% non-fat milk or 5% BSA. Fluorescents images were acquired using Biorad Chemidoc imager and HRP immunoblots were developed using standard ECL Prime (Amersham, RPN 2236).

**Real-time PCR**. Total RNA was isolated, using a Qiagen Rneasy Micro Kit, and used to produce cDNA using the Maxima first strand cDNA synthesis kit. Reverse transcription reactions were diluted to 10 ng/μl and 1–2 μl of cDNA was used to perform Real time PCR using the CFX96 real-time PCR System, CFX Manager Software and the Ssofast EvaGreen Supermix reagent. All reactions were run in triplicate and analyzed using the ΔΔCT method with relative expression normalized to *GAPDH*. (See Supplemental Table 3 for all primer sequences).

**Single cells RNA sequencing**. After tissue dissection and dissociation, FACS purified suspended cells were partitioned into nanoliter-scale Gel Bead-In-Emulsions (GEMs) with the Chromium Single Cell Controller (10× Genomics) (performed by HalioDx company and TGML Marseille Luminy platform). After cell encapsulation and barcoding, library preparation followed the standard scRNAseq protocol comprising reverse transcription, amplification, and indexing (10× Genomics). Sequencing was performed using a NextSeq Illumina device (Illumina). Illumina bcl files were basecalled, demultiplexed and aligned to the mouse mm10 genome using the cellranger software (version 3.1.0, 10× Genomics). All downstream analyses were done with R/Bioconductor packages, R version 4.0.3 (2020-10-10) [https://cran.r-project.org/; http://www.bioconductor.org/]. Raw counts were imported into R and single cell data was analyzed with the 'Seurat' package v.3.2.2[21]. After filtering for library size (between 200 and 5000 features per cell) and mitochondrial gene expression (less than 25%), pre-processing was performed using Seurat functions for counts normalization (NormalizeData using the LogNormalize method), scaling (ScaleData using the 2000 most variable features), dimension reduction with principal component analysis (PCA) (RunPCA with default parameters), construction of a shared nearest neighbor (SNN) graph (FindNeighbors using ten dimensions of reduction as input) and clustering (FindClustering with a resolution of 0.2). Data was adjusted for cell cycle using the CellCycleScoring function and regressing for S and G2/M phases scores. After visualization with Uniform Manifold Approximation and Projection (UMAP) dimensional reduction technique [https://arxiv.org/abs/1802.03426], contaminating cell types were filtered out (i.e., anal glands, hair follicle, and fibroblasts) followed by re-scaling of each dataset. An additional filtering based on the expression of *Lgr5*, *Lhx2*, *Tbx1*, was necessary in the normal sample to remove hair follicle cells that were contaminating the basal cluster. After filtering of bad quality and contaminating cells, 939 cells from the normal sample and 4147 cells from the wounded sample were used for all downstream analyses. Markers of each cluster were identified using the wilcox test option of the FindAllMarkers function, with a bonferroni *p* value adjustment. The same criteria for filtering, dimension reduction, clustering, visualization, adjusting for cell cycle, and pseudotime were used to analyze the normal and the wounded sample. Cell-filtering parameters are nFeature_RNA > 200, nFeature_RNA < 5000, percent.mt < 25%. Other parameters included ScaleData (x, vars.to.regress = c("S.Score", "G2M.Score")), RunPCA(x) # default parameters, FindNeighbors(dims = 1:10), FindClusters(resolution = 0.2), RunUMAP(dims = 1:10).

**Pseudotime analysis**. We used the dynverse R package to compare across pseudotime algorithms and choose the one best suited for our dataset[27]. Final trajectory (pseudotime) analyses were performed using the slingshot R package[26]. To this end, clustering information was extracted from Seurat objects and passed directly to slingshot's main function, using the "Basal" cell cluster as the starting cluster. The same "granularity" parameters (i.e., Omega = 20) were used for both conditions, normal and wounded, to assure comparability. Once pseudotime trajectories were identified, a general additive model (GAM) was fitted to identify genes, whose expression was significantly associated to each trajectory.

**Rectal wound**

*Chemical wound*. To remove rectal crypts close to the anorectal TZ we treated the anorectal region with hot EDTA followed by a mechanical wound as previously described[41] using 2–4 months old *K17CreER*^T2^;*R26R*^GFP^ bigenic mice injected with tamoxifen 2 days before the wound (for lineage tracing experiments). Briefly, under isofluorane anesthesia, a thin catheter equipped with a small balloon, was inserted into the anus and 200 µl of hot (50 °C) fresh 0.25 M EDTA/PBS was added using a syringe and a thin catheter. After 2 min of exposure, one wash with PBS was done and the balloon was deflated and catheters removed. Epithelial wound was performed using an electric brush closed to the anorectal TZ for approximately 1 min, and removal of the rectum crypts confirmed under a dissecting microscope. Mice were monitored carefully after the injury and analyzed 2 weeks and 4 weeks post-wound.

*Mechanical wound*. Under isofluorane anesthesia, 2–4-months-old K17CreER^T2^; R26R^GFP^ bigenic mice injected with tamoxifen 2 days before the wound, were subjected to a mechanical wound using a sterile scalpel (InterFocus #10073-14) inserted into the anus. The glandular region close to the anal transition zone was scraped with the scalpel to create the wound. Mice were monitored carefully after the injury and analyzed 1 week of post-wound.

All mice were injected subcutaneously with 50 µl of METACAM (1 mg/kg) before every wounding experiment.

**Quantifications**. Krt17 protein expression was evaluated for each region (anal canal, TZ, and rectum). At least eight areas have been quantified per regions and per mice (*n* = 3 mice per condition). Immunostainings were performed the same day with the same antibody solution and identical imaging settings were applied, when comparing different conditions. Images were acquired with a Confocal microscope ZEISS LSM 880 and were analyzed with ImageJ. A threshold was applied to all images to remove the background before measuring the integrated density, while keeping the same ROI for each images (see Supplemental methods

for details). For EdU quantification, eight areas per mice and per region were manually counted using Zen software and photoshop CC2019. Krt17 expression was used to separate TZ from anal canal and rectum. At least eight different areas of TZ and anal canal and 16 different crypts from nine mice were quantified (*n* = 3 per condition). Percentage of EdU+ cells was calculated according to total number of DAPI cells (reported as 100%). CD34 fraction was quantified and the percentage of CD34 was calculated according to total number of DAPI cells (reported as 100%). Quantification on EdU+ cells within the CD34+ population was quantified manually, and the percentage of EdU was calculated according to total number of CD34+ cells (reported as 100%).

**Reporting summary**. Further information on research design is available in the Nature Research Reporting Summary linked to this article.

## Data availability

All the raw scRNA sequencing data have been deposited in the Gene Expression Omnibus under Series GSE163394 (Accession # GSM4982239 and GSM4982240) Source data are provided with this paper.

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

## Acknowledgements

Processing of the slides for histology was performed by Emilie Agavnian in the experimental histopathology (ICEP) core facility at the Institut Paoli Calmettes. All organoid cultures were done in the 3D-Hub-O platform at CRCM. We thank Manon Richaud and Françoise Mallet in the Flow cytometry facility, Daniel Isnardon and Magda Rodriguez (especially for the RNAscope quantification) in the microscopic core facility at the CRCM, Charlyne Gard for the scRNA seq experiment at Aix Marseille Univ, INSERM, TAGC, TGML, Marseille, France. Ghislain Bidaut in the Cibi platform for the initial scRNAseq analysis, David Owens for the K17CreERT2 mouse lines and Marc Lopez for the anti-nectin-4 monoclonal antibody. Jean-Christophe Orsoni and Arnaud Capel in the mouse facility at the CRCM. Samuel Granjeaud for helping with statistics. This study was supported by ANR grant #ANR20-CE13-0009-01 (G.G.) and ANR-10-INBS-0009-10 (platform TAGC), partly supported by research funding from the Canceropôle Provence-Alpes-Côte d'Azur, Institut National du Cancer and Région Sud, grants from the Excellence Initiative of Aix-Marseille University A*Midex, "Investissement d'avenir" (CapoStromEx) and Inserm Plan Cancer AAP single cell (G.G. and H.H.V). L.M. is a recipient of the French ministerial research fellowship and the Ligue National Contre le Cancer fellowship.

## Author contributions

L.M. designed and performed all the experiments, analyzed the data, did all the quantifications and wrote the manuscript. V.C. performed some cryosectionning, immunostainings, wounding, shRNA experiments, western blot and analyzed the data. G.G. designed experiments, analyzed the data and wrote the manuscript. H.H.V. performed all the bioinformatics single cell analysis. A.O. performed cryosectionning, immunostainings and mouse genetics. Z.H. and J.P. performed all RNAscope experiments. F.P. and C.B-C. provided human anal samples and histological analysis. E.C.J provided histological analysis. C.G. assisted with gene ontology analysis and analyzed the data.

## Competing interests

The authors declare no competing interests.

## Additional information

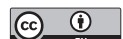

