## [Peer Review File · Nature Communications]

REVIEWER COMMENTS

Reviewer #1 (Remarks to the Author):

Mitoyan et al. have shown that Krt17+ stem cells are present in the transition zone (TZ) between the squamous anal and rectal glandular regions. By performing cell lineage tracing, organoid culture, and single cell analysis, the authors propose that they have a role in normal homeostasis and regeneration. Krt17+ cells in the TZ display multipotency, giving rise to long term progeny in anal epithelium, and possess organoid forming capacity. Upon injury, Krt17+ squamous stem cells could provide repairing progeny to the glandular epithelium. While identifying new stem cell populations in the TZ and defining their roles in adjacent tissue homeostasis and regeneration would be of general interest, some of their claims were not supported experimentally. The following major concerns must be addressed prior to consideration for publication:

Both the single cell data (Fig 3a, d) and immunofluorescence images (Fig 1a) show much more broadly labeled Krt17+ cell populations than the cells labelled by the Cre mouse line with a low dose of tamoxifen, which is a biased approach (Fig 1c), suggesting that Krt17+ doesn't specifically label basal stem cells in the TZ. Do Krt17+ cells also label basal cells in the anal epithelium? The authors should perform a comprehensive analysis of Krt17 expression to clarify this important issue.

Although Krt17 doesn't seem to label the majority of anal canal epithelial cells, the long-term lineage tracing showed that Krt17+ cells in the TZ give rise to the majority of anal epithelial cells, suggesting their potential role as anal canal stem cells (Fig 1). The authors should functionally address their role as stem cells by genetically ablating Krt17+ cells. This work would further validate the authors' claim that "Krt17+ cells maintain a squamous epithelium during normal homeostasis." Related to this point, the morphologies of organoids derived from the anal canal and TZ look similar (Fig 2b). How could organoids derived from these two tissues be distinguished? The authors also should perform long-term organoid culture derived from the TZ to examine if these organoids can give rise to anal canal epithelial cells. If so, this result would further support the role of Krt17+ cells as anal canal stem cells.

Consistent with the authors' in vivo lineage tracing data, differentiation trajectory analysis of their single cell data shows that TZ cells maintain the stratified squamous epithelium by giving rise to anal canal epithelial cells. However, the authors' designation of TZ vs anal canal cell clusters in the single cell data would still require further validation with additional, specific markers. Both Krt17 and Krt6 were claimed as TZ markers but labeled many cells in anal clusters (Fig 3a, d, e). It is still unclear if the transcriptomes of these two tissues are distinct, or if different clusters represent distinct stages in squamous epithelial differentiation. The authors mentioned that CD34 and alpha6 integrin are expressed in basal cells in the TZ, but their expression doesn't seem to be specific to the TZ (Fig 3f, g). Importantly, alpha6 integrin staining in Fig 3 and Ext data Fig 3 is very different. They should reanalyze their expression and also present the data as a tSNE plot.

This manuscript provides no clear, mechanistic insight. Using their single cell transcriptome data, the authors should investigate signaling pathways enriched in Krt17+ cell clusters.

Lineage tracing upon injury showed that Krt17+ cells also are involved in repair. However, their role as rectal epithelial stem cells is still unclear. In fact, the absence of ribbons traced from the TZ in the rectum doesn't support mobilization of Krt17+ cells from the TZ. Genetic ablation of Krt17+ cells would be able to functionally demonstrate their role in both the anal canal and rectal regeneration. Moreover, single cell analysis upon injury would provide new insight into the dynamics of Krt17+ cells in regeneration.

Reviewer #2 (Remarks to the Author):

This manuscript employs *in vivo* lineage tracing, organoid assay and single-cell transcriptional profiling approaches to identify and characterize a potentially novel population of stem-like cells residing within the transition zone between the anal canal and the rectal epithelium that contributes to maintenance of the anal canal epithelium and repair of the rectal glandular epithelium.

The study makes some novel, interesting observations, but some of the conclusions are overstated and there are some technical limitations that need to be addressed.

Major critique

1) I remain unconvinced as to the endogenous expression profile of K17 within the anal/rectal region of the mouse. According to the IF data presented in figure 1, K17 has a very narrow, highly specific expression pattern in the TZ, with nothing in the anal canal. In contrast, the transcriptomic data in fig 3 identifies a subset of K17+ cells to be anal canal cells. Also, according to the transcriptomic data, Krt17 should have a broader expression pattern than Krt6, yet by IF, the opposite appears true – Krt6 expression (Fig 3F) appears much broader/more robust than the Krt17 expression reported in Fig1. This suggests to me that there may be a disconnect between Krt17 protein localization and Krt17 transcripts. Since lineage tracing is reporting transcriptional activity of the Krt17 gene, it is therefore very important to accurately document the location of Krt17 transcripts in the anal/rectal region via high quality *in-situ* hybridization such as RNAscope. This will allow the authors to better substantiate their claim that Krt17+ cells located exclusively within the TZ zone function as the anal canal stem cells.

2) Related to the point above, it should be pointed out that 2x2mg Tamoxifen dose (the regimen reported in the methods section, but not reflected in the fig 1b schematic) is not low dose as described in the text and could be causing damage to the epithelium. Endogenous Krt17 expression in the presence/absence of tamoxifen should then also be documented to exclude the possibility that Krt17 expression is induced in the anal canal itself following damage. It would be useful to document the proliferation status of the Krt17+ TZ cells (and the postulated CD34+ stem like subset) under homeostatic and damage (post-tamoxifen and post-rectal damage) conditions to highlight any intrinsic heterogeneity and changes following damage.

3) The efficacy of the Krt17-driven lineage tracing shown in fig 1 is incredibly high (almost 100% at later time-points). This would appear to indicate that there are a very limited number of true Krt17+ stem cells and that you are somehow hitting 100% of these with your induction regime. If this is truly representative of the *in vivo* situation, it must also mean that the anal canal epithelium is very rapidly and uniformly renewed during homeostasis. Is this known? How far along the anal canal does the tracing extend? Such a rapid lateral spread of progenitor/differentiated progeny along the anal canal is difficult to reconcile with what is known from other systems. Can the authors please comment on this? Is the lineage tracing more sporadic at lower tamoxifen doses, as you would expect? Although I wouldn't expect the authors to necessarily perform this during revision, clonal lineage tracing using multicolour reporters would be useful in elucidating the relative contribution of Krt17+ cells within the TZ zone.

4) Please include control data at later time-points (> 3 weeks) to exclude longer-term leakiness in the anal canal region.

5) Fig2a – Following FACS sorting of the EPCAM+/GFP+ and cells at early time-points following tracing initiation, I would like to see q-PCR performed to prove their identity as Krt17^{high} cells.

6) Given the almost 100% tracing efficacy observed *in vivo*, I would expect a significant proportion of the Krt17+ TZ cells to demonstrate stem cell functions in the organoid assay. This should be evaluated by seeding single cells and comparing their outgrowth efficacy relative to Krt17- populations.

7) Comparison of the organoid behaviour from the different regions is only relevant if there are no regional specific *ex vivo* growth requirements. Given the distinct physiology of the different regions, it

is not unlikely that a single culture medium will be sub-optimal for one or more of the regional organoids. Given that this culture system was developed for glandular epithelia of the gut, it is perhaps not surprising that it works best to support the rectal organoids. In fig 2c, why do the anal canal organoids express Krt17 at similar levels to the TZ organoids? Shouldn't Krt17 be confined to the TZ organoids as it is in vivo? Or is this organoid system reflecting more of a regeneration scenario than a homeostatic one?

8) Why do so many of the TZ organoids crash at later passages? Does this indicate that only a subset of the TZ+ cells are true stem cells? Is there any correlation between phenotype (size, proliferation status, organization etc) and ability to be passaged long-term.

9) Given that a sub population of the anal canal-derived organoids appears to be Krt17+, does addition of 4-OH tamoxifen to the medium to initiate tracing result in clonal expansion?

10) From the single cell transcriptomic data, Krt17 is expressed in a significant number of cells classified as anal canal – could these then not be functioning as the resident stem cell population rather than the TZ cell proper? Why wasn't this observed with the in vivo lineage tracing? Perhaps they take longer to accumulate GFP expression so are missed at the early time-point presented?

11) Are the TZ cells dependent on the Wnt3a/Rspo in the culture medium – and if so, does this reflect a similar in vivo requirement? If so, it is surprising that no Wnt target genes are enriched in the TZ KRT17+ population given the plethora of Wnt genes enriched in stem cell populations along the rest of the GI tract. Is Lgr5 etc definitely absent from the TZ zone?

12) Why are HF signature genes so robustly present in the transcriptome data set? Can the authors be sure that all these genes are associated with contaminating HF cells rather than TZ cells?

13) Given that CD34, a known stem cell gene, has been identified as a marker that potentially stratifies the TZ population, why did the authors not try to sort CD34+/GFP+/EPCAM+ TZ cells at early time-points and evaluate their ability to generate organoids relative to the CD34- TZ cells? Is CD34 similarly expressed on a subset of human TZ KRT17+ cells? If so, it may be possible to evaluate their stem cell identity using the organoid assay (not a requirement for the revision process).

14) Figure 5- please clarify whether the same tamoxifen dosing regime is used in the damage and homeostatic models. This is not clear from the schematics in the respective figures. I would suggest that inducing injury only 1 day after the last Tamoxifen dose is dangerous, because if damage itself is inducing Krt17 expression in the regenerating rectal epithelium, then the remaining tamoxifen (half-life up to 2 days) will activate lineage tracing in these non TZ zone populations. I would strongly suggest to repeat with damage induction at least 3 days after the last dose of Tamoxifen. I would also like to see a Krt17 in-situ performed on the damaged rectal epithelium at early time-points following injury.

15) The frequency and location of the GFP tracing in the regenerating rectal epithelia does not convincingly identify Krt17+ TZ cells as the source of "plastic" stem cells. Why do the few GFP+ cells seen at early time-points appear in a single crypt that is several crypts distant from the TZ? Why would they not preferentially seed neighbouring crypts? How do the labelled TZ cell(s) migrate such distances? It is also surprising that the single, apparently uniformly traced crypt does not expand laterally via crypt fission as is known to aggressively occur during regeneration along the intestine. It is also essential to include an un-induced R26R-GFP/K17CreERT2 subjected to the same damage regime to rule out low-level leakiness within the damaged rectum.

16) In the concluding paragraph, the authors imply that the discovery of this "plastic" TZ stem cell pool contributing to different epithelial compartments under specific conditions is highly novel. I would argue that this is not the case since there is now a great deal of evidence for cell plasticity in many epithelia, including the intestine. What would be potentially highly novel, would be defining the role of highly plastic TZ populations in driving cancer.

Reviewer #3 (Remarks to the Author):

In this manuscript, the authors reported a novel population of Krt17+ basal cells with multipotent properties at the anorectal transition zone. Using lineage tracing and 3D organoid culture assay combined with the single-cell transcriptomics, they showed that Krt17+ cells contribute to the renewal of the squamous epithelium during homeostasis and repair the rectal glandular epithelium after injury. This work is potentially interesting. However, better evidence is needed to support the conclusions.

1. Figure 2g: TZ organoids show poorer viability compared to the rectum organoids. Does this mean the stemness of Krt17 cells is inferior to that of the rectal stem cells?
2. Figure 3: Expression of Krt17 in suprabasal cells is higher than that in basal cells, which is inconsistent with the IF result in Figure 1. Besides, 3 subtypes of cells are described in TZ. Do they exhibit any differences in gene expression profiles and in the ability to form organoids?
3. Figure 5c and 5d: Why did GFP mark a rectal crypt far from TZ, but not the ones close to TA after injury? Considering that Krt17+ cells that contribute to the self-renewal of the squamous epithelium are strictly located at TZ, it is strange to see Krt17+ cells appear far from TZ. How to explain this? Is there any possibility that Krt17 is also expressed in a subset of rectal epithelial cells, which contribute to regeneration? The possible expression of Krt17 in the rectal epithelium should be carefully examined with IF.
4. Page 6 and Figure 5d: The cells stained by alcian blue should be goblet cells, but not Paneth cells, right?
5. What is underlying mechanism of the regeneration role of the Krt17+ cells after injury? RNA-seq analysis of Krt17+ cells before and after injury should be performed. Other injury models should be used to verify the conclusion.

Summary of Our Detailed Responses to Reviewers' Suggestions and the Changes We Made to the Manuscript NCOMMS-20-09553-T and Figures:

Reviewer 1:

1) Both the single cell data (Fig 3a, d) and immunofluorescence images (Fig 1a) show much more broadly labeled Krt17+ cell populations than the cells labelled by the Cre mouse line with a low dose of tamoxifen, which is a biased approach (Fig 1c), suggesting that Krt17+ doesn't specifically label basal stem cells in the TZ. Do Krt17+ cells also label basal cells in the anal epithelium? The authors should perform a comprehensive analysis of Krt17 expression to clarify this important issue.

We have quantified the expression of Krt17 at mRNA level (by RNAscope technology, new extended Figure 2 and source data file) and at the protein level (by immunofluorescence, new extended Figure 3) and showed that Krt17 is strongly expressed in the basal cell in the TZ compared to the anal canal both at the mRNA and protein level. We show that there is a gradient, strongly in the basal cell in the TZ closer to the rectum and at lower level in the proximal part of the anal canal. This result is consistent with the Krt17 expression shown in the first submission in Figure 2e in the AC organoid versus TZ organoid where Krt17 is weakly detected in the AC organoid. We have now provided a western blot (new Figure 3d) to confirm the high Krt17 expression in the TZ organoids. Therefore the K17Cre mouse line using low dose of tamoxifen allows labeling strongly expressed Krt17+ anal TZ cells.

2) Although Krt17 doesn't seem to label the majority of anal canal epithelial cells, the long-term lineage tracing showed that Krt17+ cells in the TZ give rise to the majority of anal epithelial cells, suggesting their potential role as anal canal stem cells (Fig 1). The authors should functionally address their role as stem cells by genetically ablating Krt17+ cells. This work would further validate the authors' claim that "Krt17+ cells maintain a squamous epithelium during normal homeostasis."

We think that this approach may not give a definitive answer to the contribution of the anal TZ to the maintenance of the homeostasis because TZ plasticity of surrounding tissue may occur and recreate a new TZ as seen in the eye TZ limbus region (Nasser et al., 2018 Cell Reports). Therefore, bringing the notion of plasticity will be out of the focus of this paper. In the revised manuscript we now provide evidence that Krt17 transcript is restrict to the anal TZ and therefore the use of our lineage-tracing mouse model demonstrate clearly the contribution of the anal TZ to the maintenance of the homeostasis. This has been confirmed by our in vitro organoid assay and our single cell RNA sequencing analysis in which differentiation trajectory reconstruction show that, during normal homeostasis, anal TZ cells maintain the stratified squamous epithelium by giving rise to suprabasal TZ cells and anal canal population.

3) Related to this point, the morphologies of organoids derived from the anal canal and TZ look similar (Fig 2b). How could organoids derived from these two tissues be distinguished?

We thank the reviewer as this comment helped us to define better the anal canal population. Eventhough anal canal organoids are histologically similar to TZ organoids, they express low amount of Krt17 as show in the previous Figure 2e (now Figure 3f) (immunofluorescence) and new Figure 3d (western blot). We now show that anal canal organoids in contrast to TZ organoids express specifically the glycoprotein Gpc3 as shown in our scRNAseq analysis and confirmed by immunofluorescence on tissue and organoid (New Figures 2h and 3i).

The authors also should perform long-term organoid culture derived from the TZ to examine if these organoids can give rise to anal canal epithelial cells. If so, this result would further support the role of Krt17+ cells as anal canal stem cells.

In the first version of our manuscript long term culture has been done for 6 passages which correspond to a mean of 68 days cell growth in culture from 4 separate FACS-cell sorting and clonogenic experiments (60 days, 69 days, 82 days and 62 days). We have added this detail in the text (in Method section). To convince the reviewer that Krt17+ cells have a role as anal canal stem cells, we have extended the number of passages to 15 which correspond to 131days and obtained similar results than the one shown initially in Figure 2g. We have added this new data in the text. Moreover, TZ organoids are composed of Krt17 negative and Gpc3 positive cells showing that TZ organoids can give rise to anal canal epithelial cells. This has been added in a new Figure 3i.

4) Consistent with the authors' in vivo lineage tracing data, differentiation trajectory analysis of their single cell data shows that TZ cells maintain the stratified squamous epithelium by giving rise to anal canal epithelial cells. However, the authors' designation of TZ vs anal canal cell clusters in the single cell data would still require further validation with additional, specific markers. Both Krt17 and Krt6 were claimed as TZ markers but labeled many cells in anal clusters (Fig 3a, d, e). It is still unclear if the transcriptomes of these two tissues are distinct, or if different clusters represent distinct stages in squamous epithelial differentiation. The authors mentioned that CD34 and alpha6 integrin are expressed in basal cells in the TZ, but their expression doesn't seem to be specific to the TZ (Fig 3f, g). Importantly, alpha6 integrin staining in Fig 3 and Ext data Fig 3 is very different. They should reanalyze their expression and also present the data as a tSNE plot.

To clarify the differences between the TZ and the anal canal we have provided new immunofluorescence stainings and we have explained better the expression of each markers.

1. *Specific to the TZ are CD34 and Krt17 as shown initially in Figure 3 gii. Due to a low level of mRNA detection, CD34 cannot be presented as a UMAP.*
2. *a6-integrin has been used to mark all basal layer cells of the TZ, anal canal and rectum and we have not claimed that it is specific to the TZ, we have clarified that in the text. We now*

provided a UMAP representation of alpha6 integrin showing its broad expression in the basal layer and some suprabasal layer of the TZ clusters and the anal canal (new Figure 2e). The difference of alpha6 integrin staining in Fig 3 and Extended data Fig 3 is due to difference in fixation of the tissue in methanol in figure 3 and paraformaldehyde in Extended figure 3. We have used methanol in figure 3 as Nectin is only detected using this fixation method. We have now provided staining of perilipin with a6-integrin in methanol-fixed tissue and show no difference in alpha6 integrin staining between figures (New Extended figure 5c).

3. *Krt6 has not been claimed to be TZ markers. In the tSNE initially presented in figure 3e and summarized in the new figure 2e (shown as a UMAP), Krt6 marks the TZ and suprabasal cells of the anal canal proximal to the TZ. This is also seen in the co-staining of Krt6 and CD34 where CD34 marks the TZ (Figure 2g).*
4. *As suggested by the reviewer, to further validate our scRNA seq analysis we have found an additional marker the glycoprotein Gpc3 that is highly expressed in the AC population compared to the TZ. We have provided a UMAP representation and an immunofluorescence confirmation in vivo and in vitro in the organoid culture (New Figure 2c, 2h and 3i)*
- 5) *This manuscript provides no clear, mechanistic insight. Using their single cell transcriptome data, the authors should investigate signaling pathways enriched in Krt17+ cell clusters.*

We have analyzed pathways enriched in Krt17+ cell clusters and focused on the Jun family as interestingly this pathway has been involved in other epithelial stem cells such as hair follicle bulge stem cells and is associated with wound response. First, we have confirmed by immunostaining that the AP-1 transcription factor JunB is enriched at the anal TZ but not the anal canal or the rectum in vivo and in vitro (new figures 6d-e JunB in tissue and JunB in organoid) confirming the scRNAseq analysis. Moreover, in a wounded situation we showed that JunB is also enriched in cells participating in the wound repair (new Figure 6f). We further showed using shRNA technology in the organoid culture that JunB is essential for differentiation properties of anal TZ cells (new Figure 6i-j).

6) Lineage tracing upon injury showed that Krt17+ cells also are involved in repair. However, their role as rectal epithelial stem cells is still unclear. In fact, the absence of ribbons traced from the TZ in the rectum doesn't support mobilization of Krt17+ cells from the TZ. Genetic ablation of Krt17+ cells would be able to functionally demonstrate their role in both the anal canal and rectal regeneration. Moreover, single cell analysis upon injury would provide new insight into the dynamics of Krt17+ cells in regeneration.

In order to follow the mobilization of Krt17+ cells from the TZ after an injury we now provide earlier time point (1 week post-wound EDTA) and show GFP+ cells generating a novel rectal crypt (New figure 5c). Moreover we provide new images of later time point showing ribbon of GFP+ cells in 2 crypts (New Figure 5c). As asked by the reviewer, we have performed a new scRNA seq upon injury after 1 week post-EDTA wound to follow the dynamics of Krt17+ cells in regeneration (New Figure 6a-c and new Extended Figure 8). We have re-analyzed our single cell

RNA sequencing data in the normal condition (new figure 2, new extended figure 5 and new figure 4) with the same criteria than the new single cell RNA sequencing in the wounded condition; In the first submission we used tSNE for projection of single cell clusters in two-dimensional plots. In the revised version we visualized the clusters with Uniform Manifold Approximation and Projection (UMAP) dimensional reduction technique. Since our first submission, UMAP projections have replaced tSNE as the default representation of single cell transcriptome data due to computational efficiency and interpretability. Indeed, the cluster space in UMAP plots are better associated with biologically meaningful relationships between cell subtypes. Therefore, for comparability and reproducibility, we have replaced all tSNE projections by UMAP. We have obtained similar results for the normal condition that can be compared to the wounded condition.

This new analysis highlights a cluster of TZ cells in the wounded condition that is in a transition state and show activation of the JunB pathway. Moreover, pseudotime analysis confirmed our in vivo data showing a new trajectory of TZ wounded cells going toward rectal cells (new figure 6b-c). Concerning the genetic ablation, we think that it may not answer fully this point if surrounding tissue has the plasticity to recreate a TZ (as seen in the normal eye) and therefore opening another question out of the scope of this paper.

Reviewer 2:

1) I remain unconvinced as to the endogenous expression profile of K17 within the anal/rectal region of the mouse. According to the IF data presented in figure 1, K17 has a very narrow, highly specific expression pattern in the TZ, with nothing in the anal canal. In contrast, the transcriptomic data in fig 3 identifies a subset of K17+ cells to be anal canal cells. Also, according to the transcriptomic data, Krt17 should have a broader expression pattern than Krt6, yet by IF, the opposite appears true – Krt6 expression (Fig 3F) appears much broader/more robust than the Krt17 expression reported in Fig1. This suggests to me that there may be a disconnect between Krt17 protein localization and Krt17 transcripts. Since lineage tracing is reporting transcriptional activity of the Krt17 gene, it is therefore very important to accurately document the location of Krt17 transcripts in the anal/rectal region via high quality in-situ hybridization such as RNAscope. This will allow the authors to better substantiate their claim that Krt17+ cells located exclusively within the TZ zone function as the anal canal stem cells.

As suggested by the reviewer we have performed high quality in-situ hybridization RNAscope to detect Krt17 transcripts in the anal/rectal region. The quantification and images show that Krt17 transcript is restricted to the anal TZ and not the anal canal (see representative example in a new Extended figure 2 and all the quantification data are available in a new data availability section)

2) Related to the point above, it should be pointed out that 2x2mg Tamoxifen dose (the regimen reported in the methods section, but not reflected in the fig 1b schematic) is not low dose as described in the text and could be causing damage to the epithelium. Endogenous Krt17 expression in the presence/absence of tamoxifen should then also be documented to exclude the possibility that Krt17 expression is induced in the anal canal itself following damage.

We have used two protocol of Tamoxifen injection. In figure 1, we have used only one injection of 2 mg of Tamoxifen (as presented in Figure 1b) to only label few cells at the TZ and follow their potential to give rise to anal canal cells. We have added this important detail in the figure legend. In the next figures we have used 2x2mg Tamoxifen dose to label all cells in the anal TZ in order to isolate them (Figure 2) and follow their potential to repair a wound (Figure 5). We have removed low dose from the text when we used 2x2mg of Tamoxifen and replace low dose of tamoxifen by one dose of tamoxifen when we describe the figure 1.

As requested by the reviewer, we have now quantified the expression of Krt17 by immunofluorescence in the presence/absence of tamoxifen and after 48h post-wounding to verify that Tamoxifen treatment itself does not induce a wound response. We have also monitored the expression of the Edu to mark proliferative cells (see point 3 below). These new data presented in Extended figure 3 show that Tamoxifen treatment does not induce significantly Krt17 expression in the anal canal and does not resemble a post wounding condition when we compared to a 48h post-wound condition (New extended figure 3).

3) It would be useful to document the proliferation status of the Krt17+ TZ cells (and the postulated CD34+ stem like subset) under homeostatic and damage (post-tamoxifen and post-rectal damage) conditions to highlight any intrinsic heterogeneity and changes following damage.

We have monitored the expression of Edu to mark proliferative cells and performed immunofluorescence with Krt17 and CD34 in 3 conditions: without TAM injection, in 2 days TAM injected mice and in 2 days TAM and 48h post-rectal wound. These data have been incorporated in a new Extended figure 3. Quantifications show that TAM injection itself does not change the proliferation status of K17+ TZ cells, neither the expression of CD34 at the TZ.

4) The efficacy of the Krt17-driven lineage tracing shown in fig 1 is incredibly high (almost 100% at later time-points). This would appear to indicate that there are a very limited number of true Krt17+ stem cells and that you are somehow hitting 100% of these with your induction regime. If this is truly representative of the in vivo situation, it must also mean that the anal canal epithelium is very rapidly and uniformly renewed during homeostasis. Is this known?

We have provided a new image that represents better the mosaicism we see in the long-term lineage tracing after 1 injection (new figure 1d after one year and new extended figure 1c after 84 days). The renewing time of the anal canal has not been reported. According to our lineage tracing data, the anal region seems a highly regenerative; this seems consistent with the fact that this region is subject to frequent activity such as stool passage.

5) How far along the anal canal does the tracing extend? Such a rapid lateral spread of progenitor/differentiated progeny along the anal canal is difficult to reconcile with what is known from other systems. Can the authors please comment on this? Is the lineage tracing more sporadic at lower tamoxifen doses, as you would expect? Although I wouldn't expect the authors to necessarily perform this during revision, clonal lineage tracing using multicolour reporters would be useful in elucidating the relative contribution of Krt17+ cells within the TZ zone.

The lineage tracing of the anal TZ reaches the entire anal canal and stops at the perianal skin. The anal region seems a highly regenerative tissue according to the lineage tracing data; this seems consistent with the fact that this region is subject to frequent activity such as stool passage. Lower tamoxifen doses (1mg) is in fact more sporadic.

6) Please include control data at later time-points (> 3 weeks) to exclude longer-term leakiness in the anal canal region.

In figure 1 we have added a 4 weeks oil control to exclude longer-term leakiness in the anal canal region. We have put the Day 2 oil control in Extended figure 1.

7) Fig2a – Following FACS sorting of the EPCAM+/GFP+ and cells at early time-points following tracing initiation, I would like to see q-PCR performed to prove their identity as Krt17high cells.

In new figure 3d we have provided a western blot showing the identity of the anal canal EPCAM+/GFP-, TZ EPCAM+/GFP+ and rectum EPCAMhigh/GFP- organoids derived from FACS sorted cells 2 days after tracing with Krt17 and Krt8 antibodies.

8) Given the almost 100% tracing efficacy observed in vivo, I would expect a significant proportion of the Krt17+ TZ cells to demonstrate stem cell functions in the organoid assay. This should be evaluated by seeding single cells and comparing their outgrowth efficacy relative to Krt17- populations.

Here the reviewer points out that in the initial Figure 2g not all TZ clone give rise to clones as seen in the rectum. To explain this result we have showed that our initial FACS sorting strategy includes an heterogeneous anal TZ population containing basal (GFP+Nectin-) and suprabasal (GFP+Nectin+) populations. We have now sorted these populations and show that GFP+Nectin- are enriched for TZ basal cells are the one that are more clonogenic and explain why we have not 100% clonogenicity in organoids (New Extended Figure 6c-d).

9) Comparison of the organoid behaviour from the different regions is only relevant if there are no regional specific ex vivo growth requirements. Given the distinct physiology of the different regions, it is not unlikely that a single culture medium will be sub-optimal for one or more of the regional organoids. Given that this culture system was developed for glandular epithelia of the gut, it is perhaps not surprising that it works best to support the rectal organoids.

We have answered this comment in point 8.

In fig 2c, why do the anal canal organoids express Krt17 at similar levels to the TZ organoids? Shouldn't Krt17 be confined to the TZ organoids as it is in vivo? Or is this organoid system reflecting more of a regeneration scenario than a homeostatic one?

We have now provided a western blot analysis showing the expression of Krt17 in AC organoid

versus TZ organoid (new figure 3d) and show that they do not express similar level than anal TZ organoids. The low level of Krt17 seen in vitro may reflect culture condition or may come from the gradient of Krt17 in the AC from proximal to distal of the TZ.

10) Why do so many of the TZ organoids crash at later passages? Does this indicate that only a subset of the TZ+ cells are true stem cells? Is there any correlation between phenotype (size, proliferation status, organization etc) and ability to be passaged long-term.

We have answered this issue in point 8. Here the reviewer points out that in the initial Figure 2g not all TZ clone give rise to clones as seen in the rectum. To explain this result we have showed that our initial FACS sorting strategy includes an heterogeneous anal TZ population containing basal (GFP+Nectin-) and suprabasal (GFP+Nectin+) populations. We have now sorted these populations and show that GFP+Nectin-are enriched for TZ basal cells are the one that are more clonogenic and explain why we have not 100% clonogenicity in organoids (New Extended Figure 6b-d).

11) Given that a sub population of the anal canal-derived organoids appears to be Krt17+, does addition of 4-OH tamoxifen to the medium to initiate tracing result in clonal expansion?

We have added TAM in the AC organoid culture media and show that it never expresses GFP, suggesting that the lineage tracing is really specific to the TZ population (new Extended figure 6a).

12) From the single cell transcriptomic data, Krt17 is expressed in a significant number of cells classified as anal canal – could these then not be functioning as the resident stem cell population rather than the TZ cell proper? Why wasn't this observed with the in vivo lineage tracing? Perhaps they take longer to accumulate GFP expression so are missed at the early time-point presented?

AC cells expressed low level of Krt17 quantified by RNAscope technology and by immunofluorescence. Moreover, adding TAM in the AC organoid culture media never induces lineage tracing suggesting that we can not miss GFP expression due to a lack of accumulation. There is a gradient of Krt17 in the AC from proximal to distal of the TZ.

13) Are the TZ cells dependent on the Wnt3a/Rspo in the culture medium – and if so, does this reflect a similar in vivo requirement? If so, it is surprising that no Wnt target genes are enriched in the TZ KRT17+ population given the plethora of Wnt genes enriched in stem cell populations along the rest of the GI tract. Is Lgr5 etc definitely absent from the TZ zone?

scRNA sequencing analysis of the TZ population does not show significantly WNT pathway suggesting that other signaling pathway are required in this TZ niche. Consistent with this data and to answer the reviewer point 13, we now show by RNAscope technology that LGR5 is expressed in the rectal crypt as expected but not at the anal TZ (new Extended figure 5g).

14) Why are HF signature genes so robustly present in the transcriptome data set? Can the

authors be sure that all these genes are associated with contaminating HF cells rather than TZ cells?

Eventhough we microdissect the anorectal region we cannot exclude to have hair follicle contamination in our cell extraction strategy. Moreover, we have confirmed by immunofluorescence that hair follicle genes such as Lhx2 is not expressed in the TZ (New extended figure 5f).

15) Given that CD34, a known stem cell gene, has been identified as a marker than potentially stratifies the TZ population, why did the authors not try to sort CD34+/GFP+/EPCAM+ TZ cells at early time-points and evaluate their ability to generate organoids relative to the CD34- TZ cells? Is CD34 similarly expressed on a subset of human TZ KRT17+ cells? If so, it may be possible to evaluate their stem cell identity using the organoid assay (not a requirement for the revision process).

As suggested by the reviewer, we have sorted anal TZ CD34+ and anal TZ CD34- population to generate organoids. Reproducibly, anal TZ CD34+ sorted cells grow organoids more than 7 passages. However, we have obtained a lot of variability within 3 experiments when we grow anal TZ CD34- sorted cells. We explained this result by the fact that in some organoids initially CD34neg, CD34 get re-expressed in vitro and therefore preventing us to clearly answer the role of CD34 as a marker to enrich TZ for stemness properties.

CD34 is not expressed in anal TZ human sample (see image showing expression only in endothelial cells surrounding the TZ);

16) Figure 5- please clarify whether the same tamoxifen dosing regime is used in the damage and homeostatic models. This is not clear from the schematics in the respective figures.

We have clarified in the text the tamoxifen injection strategy. We have used two protocol of Tamoxifen injection. In homeostatic model (figure 1), we have used only one injection of 2 mg of Tamoxifen (as presented in Figure 1b) to only label few cells at the TZ and follow their potential to give rise to anal canal cells. We have added this important detail in the figure legend that was missing. In the damage models (Figure 5) we have used 2x2mg Tamoxifen dose to label all cells in the anal TZ in order to follow their potential to repair a wound.

17) I would suggest that inducing injury only 1 day after the last Tamoxifen dose is dangerous,

because if damage itself is inducing Krt17 expression in the regenerating rectal epithelium, then the remaining tamoxifen (half-life up to 2 days) will activate lineage tracing in these non TZ zone populations. I would strongly suggest to repeat with damage induction at least 3 days after the last dose of Tamoxifen. I would also like to see a Krt17 in-situ performed on the damaged rectal epithelium at early time-points following injury.

We have now provided all controls to show that there is no possible expression of Krt17 in the rectal epithelium. We have performed quantification at the protein level and at the mRNA level (by RNAscope technology) in the presence/absence of tamoxifen and after 1 week post-wounding to show that Krt17 itself is never expressed in the rectum. These new data are presented in Extended figure 2-3

18) The frequency and location of the GFP tracing in the regenerating rectal epithelia does not convincingly identify Krt17+ TZ cells as the source of “plastic” stem cells. Why do the few GFP+ cells seen at early time-points appear in a single crypt that is several crypts distant from the TZ? Why would they not preferentially seed neighbouring crypts? How do the labelled TZ cell(s) migrate such distances? I think it's not migration but these cells are pushed away by the surrounding crypts that also participates in repair (Show with schema) It is also surprising that the single, apparently uniformly traced crypt does not expand laterally via crypt fission as is known to aggressively occur during regeneration along the intestine

The wound we performed will activate two stem cell compartment (the rectal stem cell and the anal TZ). We think that because of this competition, it is not surprising that we do not obtain similar regeneration process as seen in the intestine when only intestinal stem cells are traced. We have analyzed early time point after the wound (1 week) and show now that we can detect anal TZ GFP cells at the TZ that expand laterally and are regenerating a crypt. We have also provided a new later time point where we can detect ribbon of rectal crypt (new Figure 5c). To also confirm the TZ response to wounding we have taken a silico approach by analyzing the trajectories taken by TZ cells after a wound. These new single cell data are now presented in Figure 6a-c and in Extended figure 8 and identified a group of TZ cells in a transition state reflecting our lineage tracing in vivo.

It is also essential to include an un-induced R26R-GFP/K17CreERT2 subjected to the same damage regime to rule of low-level leakiness within the damaged rectum.

We have provided an oil control after 4 weeks post-wound and never detect any leakiness within the damaged rectum, (new Figure 5c).

19) In the concluding paragraph, the authors imply that the discovery of this “plastic” TZ stem cell pool contributing to different epithelial compartments under specific conditions is highly novel. I would argue that this is not the case since there is now a great deal of evidence for cell plasticity in many epithelia, including the intestine. What would be potentially highly novel, would be defining the role of highly plastic TZ populations in driving cancer.

As suggested by the reviewer, we have changed this statement and added that " Understanding how these TZ cells participate in driving cancer will be highly relevant for cancer prevention and regression".

Reviewer 3:

1) Figure 2g: TZ organoids show poorer viability compared to the rectum organoids. Does this mean the stemness of Krt17 cells is inferior to that of the rectal stem cells?

To explain this result we have showed that our initial FACS sorting strategy includes an heterogeneous anal TZ population containing basal (GFP+Nectin-) and suprabasal (GFP+Nectin+) populations. We have now sorted these populations and show that GFP+Nectin- are enriched for TZ basal cells are the one that are more clonogenic and explain why we have not 100% clonogenicity in organoids (New Extended Figure 6c-d).

2) Figure 3: Expression of Krt17 in suprabasal cells is higher than that in basal cells, which is inconsistent with the IF result in Figure 1.

It is possible to have a difference in mRNA level versus protein level.

Besides, 3 subtypes of cells are described in TZ. Do they exhibit any differences in gene expression profiles and in the ability to form organoids?

We have answered this remark in point 1 above. To explain this result we have showed that our initial FACS sorting strategy includes an heterogeneous anal TZ population containing basal (GFP+Nectin-) and suprabasal (GFP+Nectin+) populations. We have now sorted these populations and show that GFP+Nectin- are enriched for TZ basal cells are the one that are more clonogenic and explain why we have not 100% clonogenicity in organoids (New Extended Figure 6c-d).

We have also sorted anal TZ CD34+ and anal TZ CD34- population to generate organoids. Reproducibly, anal TZ CD34+ sorted cells grow organoids more than 7 passages. However, we have obtained a lot of variability within 3 experiments when we grow anal TZ CD34- sorted cells. We explained this result by the fact that in some organoids initially CD34neg, CD34 get re-expressed in vitro and therefore preventing us to clearly answer the role of CD34 as a marker to enrich TZ for stemness properties.

3) Figure 5c and 5d: Why did GFP mark a rectal crypt far from TZ, but not the ones close to TA after injury? Considering that Krt17+ cells that contribute to the self-renewal of the squamous epithelium are strictly located at TZ, it is strange to see Krt17+ cells appear far from TZ. How to explain this? Is there any possibility that Krt17 is also expressed in a subset of rectal epithelial cells, which contribute to regeneration? The possible expression of Krt17 in the rectal epithelium

should be carefully examined with IF.

We have now provided all controls to show that there is no possible expression of Krt17 in the rectal epithelium. We have performed quantification at the protein level and at the mRNA level (by RNAscope technology) in the presence/absence of tamoxifen and after 1 week post-wounding to show that Krt17 itself is never expressed in the rectum. These new data are presented in Extended figure 2-3

To explain why we rarely found GFP+ TZ-derived crypt closed to the anal TZ after wounding, we discussed the fact that the wound we performed will activate two stem cell compartment (the rectal stem cell and the anal TZ). We think that because of this competition, it is not surprising that we do not obtain similar regeneration process as seen in the intestine when only intestinal stem cells are traced. We have analyzed early time point after the wound (1 week) and show now that we can detect anal TZ GFP cells at the TZ that expand laterally and are regenerating a crypt. We have also provided a new later time point where we can detect ribbon of rectal crypt (New figure 5c). Finally, to also confirm the TZ response to wounding we have taken a silico approach by analyzing the trajectories taken by TZ cells after a wound. These new single cell data are now presented in Figure 6 a-c and in Extended figure 8 and identified a group of TZ cells in a transition state reflecting our lineage tracing in vivo.

4) Page 6 and Figure 5d: The cells stained by alcian blue should be goblet cells, but not Paneth cells, right?

We have changed this mistake in the legend of the figure

5) What is underlying mechanism of the regeneration role of the Krt17+ cells after injury? RNA-seq analysis of Krt17+ cells before and after injury should be performed.

We have now performed a new single cell RNA seq after injury (new figure 6a, extended figure 8) We have analyzed pathways enriched in Krt17+ cell clusters and focused on the Jun family as interestingly this pathway has been involved in other epithelial stem cells such as hair follicle bulge stem cells and is associated with wound response. First, we have confirmed by immunostaining that the AP-1 transcription factor JunB is enriched at the anal TZ but not the anal canal or the rectum in vivo and in vitro (new figures 6d-e JunB in tissue and JunB in organoid) confirming the scRNAseq analysis. Moreover, in a wounded situation we showed that JunB is also enriched in cells participating in the wound repair (new Figure 6f). We further showed using shRNA technology in the organoid culture that JunB is essential for differentiation properties of anal TZ cells (new Figure 6i-j).

Other injury models should be used to verify the conclusion.

We have provided another method of injury which is mechanic (instead of chemical) where we injured the anal TZ with a scalpel. We show similar result than the one initially presented with

EDTA treatment. We provide now this new data in Extended Fig 7 where we show similar contribution of GFP+ anal TZ cells to the rectal regeneration and involvement of the JunB pathway in the regenerative crypt. We have added this procedure, covered in our approved animal protocol #8287, in the material and methods section.

REVIEWER COMMENTS

Reviewer #1 (Remarks to the Author):

Although the authors did not address some of my major concerns addressing the in vivo functional role of Krt17+ cells in homeostasis and regeneration, their new single cell analysis of regeneration has identified "a TZ hybrid state," which supports multipotency induced by TZ cells shown in Fig 5. This new analysis also has identified JunB, the expression of which is increased in the cells participating in wound repair. By knocking down this gene in organoid culture, the authors have shown its role in epithelial differentiation. Therefore, this revision has significantly improved the manuscript, providing some new mechanistic insight into regeneration. However, I am still concerned about some key validations.

Regarding my major concern #1, the authors performed the revision experiments and stated, "We showed that Krt17 transcript was exclusively found in the anal TZ region when compared to the anal canal and the rectum (Extended Data Fig.2a-b and quantified in Extended Fig. 2c)." However, their new data and this claim still conflict with the Krt17 UMAP plot in Fig 2d, which clearly shows the expression of Krt17 in the anal canal cluster cells. This requires clarification and should be addressed/explained in the revised manuscript.

Regarding my major concern #3, the authors stated "Each sorted cell population, stratified (GFP-EpCamLow), TZ (GFP+EpCamLow) and glandular (GFP-EpCamHigh) can grow as organoid culture and resemble their tissue of origin by histology (Fig. 3b) and molecularly by qPCR (Fig. 3c), Western blot (Fig. 3d) and immunofluorescence (Fig. 3e-f)." The authors should still explain further how histology can help distinguish their tissue origins. The images in Fig 3b are very difficult to understand. Higher magnification images could be helpful. Tissue specific morphologies and cellular features should be described.

The authors stated "Moreover, in a wounded situation we showed that JunB is also enriched in cells participating in both chemical and mechanical wound repair (Fig.6f and Ext. Data Fig. 7c)." The current Fig 6c focused only on the 4-6 clusters. The authors should also include other epithelial clusters to further validate their claim at the transcriptional level.

Reviewer #2 (Remarks to the Author):

I am still not convinced by the explanation provided for the appearance of lineage tracing units so far distant from the proposed stem cell population following damage. Is there any additional experimentation that can be performed to better substantiate this model?

Reviewer #3 (Remarks to the Author):

My concerns have been addressed. No more questions.

Summary of Our Detailed Responses to Reviewers' Suggestions and the Changes We Made to the Manuscript NCOMMS-20-09553A and Figures:

Reviewer #1 (Remarks to the Author):

Although the authors did not address some of my major concerns addressing the in vivo functional role of Krt17+ cells in homeostasis and regeneration, their new single cell analysis of regeneration has identified “a TZ hybrid state,” which supports multipotency induced by TZ cells shown in Fig 5. This new analysis also has identified JunB, the expression of which is increased in the cells participating in wound repair. By knocking down this gene in organoid culture, the authors have shown its role in epithelial differentiation. Therefore, this revision has significantly improved the manuscript, providing some new mechanistic insight into regeneration. However, I am still concerned about some key validations.

Regarding my major concern #1, the authors performed the revision experiments and stated, “We showed that Krt17 transcript was exclusively found in the anal TZ region when compared to the anal canal and the rectum (Extended Data Fig.2a-b and quantified in Extended Fig. 2c).” However, their new data and this claim still conflict with the Krt17 UMAP plot in Fig 2d, which clearly shows the expression of Krt17 in the anal canal cluster cells. This requires clarification and should be addressed/explained in the revised manuscript.

In the first revision of the manuscript, we have mentioned in the response to this concern that we show that there is a gradient of Krt17 expression strongly in the TZ cells and at a lower level in the proximal part of the anal canal. We apologize that we did not clearly mention it in the text. In this second revised version, we have included the notion of gradient in the text (page 3 and Extended data Fig.1b) showing the Krt17 gradient, strongly in the TZ cells and at lower level in the proximal part of the anal canal.

RNAscope technology is less sensitive than the single cell analysis so it is not surprising to have only detected the strong expression of the Krt17mRNA expression in the TZ. Therefore, the RNAscope data does not conflict the Krt17 UMAP plot in Fig. 2d. We have now clarified this issue in the text (page 5) by explaining that in the UMAP the mRNA Krt17 weakly positive in the anal canal cluster cells corresponds to the proximal cells of the anal canal close to the TZ and the mRNA Kr17neg in the anal canal cluster corresponds to the distal cells of the anal canal. In the first revision we confirmed that GFP was only induced in TZ organoids and not in anal canal organoids (Extended Data Fig.6a).

Regarding my major concern #3, the authors stated “Each sorted cell population, stratified (GFP-EpCamLow), TZ (GFP+EpCamLow) and glandular (GFP-EpCamHigh) can grow as organoid culture and resemble their tissue of origin by histology (Fig. 3b) and molecularly by qPCR (Fig. 3c), Western blot (Fig. 3d) and immunofluorescence (Fig. 3e-f).” The authors should still explain further how histology can help distinguish their tissue origins. The images in Fig 3b are very difficult to understand. Higher magnification images could be helpful. Tissue specific

morphologies and cellular features should be described

We have better explained in the text (page 5-6) that histology can help distinguishing squamous and glandular origin only. We have provided zoom into the histology Fig 3b showing a multilayer epithelium in the squamous TZ and anal canal organoids and a single layer of columnar cells in the glandular rectal organoid. To differentiate molecularly between TZ and anal canal we have used the molecular marker Gpc3 as already mentioned in the first revised version of the manuscript. In page 6 we have now better explained that Gpc3 is found in the anal canal organoid (Fig. 3f) whereas in the TZ, Gpc3 is found in some regions of the organoid (Fig. 3i) showing the multilineage potential of the Krt17+ cells as previously demonstrated in vivo.

The authors stated "Moreover, in a wounded situation we showed that JunB is also enriched in cells participating in both chemical and mechanical wound repair (Fig.6f and Ext. Data Fig. 7c)." The current Fig 6c focused only on the 4-6 clusters. The authors should also include other epithelial clusters to further validate their claim at the transcriptional level.

In the first revision of the manuscript we focused on the 4-6 clusters to show an enrichment of JunB in the TZ hybrid. As requested by the reviewer, we now provided the full UMAP JunB in wound in Extended Fig 8e where it not surprising to see JunB highly expressed in the cluster "epithelial cell in regeneration" as JunB is known to be overexpressed in wound healing. We have changed "enriched" by "expressed" in page 10.

Reviewer #2 (Remarks to the Author):

I am still not convinced by the explanation provided for the appearance of lineage tracing units so far distant from the proposed stem cell population following damage. Is there any additional experimentation that can be performed to better substantiate this model?

In the first revision of the manuscript we showed that at 1 week post-wound (which corresponds to an early phase of repair), few TZ cells display a phenotypic plasticity and expand laterally (Fig.5c, Extended Fig 7c, validated with two methods of wound). This has been confirmed by a scRNA sequencing analysis. At later time point we show a ribbon of rectal crypt (Figure 5c) derived from the Krt17+ TZ cells that is distant from the TZ region. We agree that this point should be better explained and interpreted. For that we have:

1. Better described the lineage tracing data after wounding in page 8 by writing that "the GFP+ rectal crypts derived from the Krt17+ TZ cells are not found continuous to the TZ such as seen when colonic stem cells are lineage traced after injury³¹. We postulate that the wound activates two stem cell compartments (the rectal stem cell and the anal TZ) and rectal cells are dominantly responsible for the regeneration."

2. Analyzed the proliferation status of rectal cells and TZ cells after wounding. Staining with Edu at shorter time point after the wound (Ext Fig 3b 48h) shows that rectal cells are highly proliferative compared to TZ. This could suggest that residual rectal epithelial cells closer to TZ rapidly expand and it could contribute to the distance between anal Krt17+ TZ cell tracing and TZ

over time. This evidence could explain distant anal Krt17+ TZ cell tracing at later time points. We have added this explanation in the text in page 8.

3. Revised our claim in Pages 8 and 10 that "anal Krt17+ TZ cells are tuned to tissue repair" and replaced by "a minority of anal Krt17+ TZ cells can contribute to rectal repair". We have also removed the sentence in page 2 in the introduction "The findings challenge the idea that each epithelium may be maintained by its own stem cell pool and open novel way for tissue regeneration following rectal injury".

4. We have now focused the discussion on the phenotypic plasticity of the TZ cells after wounding and the role in tumor initiation, instead of their regenerative capacity. Page 11: "Instead, here we propose that TZ cells could represent a backup reservoir of stem cells for surrounding epithelium and their phenotypic plasticity seen during wound healing could represent an hallmark of tumor initiation³⁷." "In human and mouse, TZ are cancer-prone stem cell niches. Therefore, understanding how these TZ cells participate in driving cancer will be highly relevant for cancer prevention and regression".

*Reviewer #3 (Remarks to the Author):
My concerns have been addressed. No more questions.*

REVIEWERS' COMMENTS

Reviewer #1 (Remarks to the Author):

My concerns have been addressed.

Reviewer #2 (Remarks to the Author):

No further comments